# HOPE FOR A ROBUST PARAMETERIZATION OF LONG-MEMORY STATE SPACE MODELS

**Annan Yu**[1]  **Michael W. Mahoney**[2-4]  **N. Benjamin Erichson**[2,3]

[1] Center for Applied Mathematics, Cornell University, Ithaca, NY 14853, USA

[2] Lawrence Berkeley National Laboratory, Berkeley, CA 94720, USA

[3] International Computer Science Institute, Berkeley, CA 94704, USA

[4] Department of Statistics, University of California at Berkeley, Berkeley, CA 94720, USA

`ay262@cornell.edu`  `mmahoney@stat.berkeley.edu`  `erichson@icsi.berkeley.edu`

## ABSTRACT

State-space models (SSMs) that utilize linear, time-invariant (LTI) systems are known for their effectiveness in learning long sequences. To achieve state-of-the-art performance, an SSM often needs a specifically designed initialization, and the training of state matrices is on a logarithmic scale with a very small learning rate. To understand these choices from a unified perspective, we view SSMs through the lens of Hankel operator theory. Building upon it, we develop a new parameterization scheme, called HOPE, for LTI systems that utilizes Markov parameters within Hankel operators. Our approach helps improve the initialization and training stability, leading to a more robust parameterization. We efficiently implement these innovations by nonuniformly sampling the transfer functions of LTI systems, and they require fewer parameters compared to canonical SSMs. When benchmarked against HiPPO-initialized models such as S4 and S4D, an SSM parameterized by Hankel operators demonstrates improved performance on Long-Range Arena (LRA) tasks. Moreover, our new parameterization endows the SSM with non-decaying memory within a fixed time window, which is empirically corroborated by a sequential CIFAR-10 task with padded noise.

## 1 INTRODUCTION

State-space models (SSMs) (Gu et al., 2022b) have gained popularity and success in sequence modeling. Known for its excellent efficiency and capability of handling long sequences, an SSM leverages the continuous-time linear, time-invariant (LTI) systems. These systems are often defined by four matrices $\Gamma = (\mathbf{A}, \mathbf{B}, \mathbf{C}, \mathbf{D})$ as

$$\mathbf{x}'(t) = \mathbf{A}\mathbf{x}(t) + \mathbf{B}\mathbf{u}(t), \qquad \mathbf{y}(t) = \mathbf{C}\mathbf{x}(t) + \mathbf{D}\mathbf{u}(t), \tag{1}$$

and they can be used to model the mappings from input time-series $\mathbf{u}(\cdot)$ to the output times-series $\mathbf{y}(\cdot)$, where $\mathbf{u}(t) \in \mathbb{R}^m$ and $\mathbf{y}(t) \in \mathbb{R}^p$ for every $t$. The (hidden) states, which capture the latent dynamics, are denoted as $\mathbf{x} = \mathbf{x}(t) \in \mathbb{R}^n$. The system matrices are of dimensions $\mathbf{A} \in \mathbb{C}^{n \times n}$, $\mathbf{B} \in \mathbb{C}^{n \times m}$, $\mathbf{C} \in \mathbb{C}^{p \times n}$, and $\mathbf{D} \in \mathbb{C}^{p \times m}$. Often, the size $n$ of the state vector $\mathbf{x}$ is much larger than $m$ and $p$, which allows us to memorize information about the past inputs $\mathbf{u}|_{(-\infty,t]}$ in the state vector $\mathbf{x}(t)$ and retrieve it later to compute $\mathbf{y}$ via $\mathbf{C}$.

The so-called S4 (Gu et al., 2022b) and S4D (Gu et al., 2022a) models both set $m = p = 1$, and they differ in the structural requirement of $\mathbf{A}$. This framework was later generalized to the case where $m, p > 1$ by the S5 model (Smith et al., 2023) via the parallel scans. Another line of research involves making the state transition rule $\mathbf{A}$ depend on the input $\mathbf{u}$, along which the two most notable models are Liquid-S4 (Hasani et al., 2023) and Mamba (Gu & Dao, 2023), where the latter model achieves the state-of-the-art performance on large-scale real-world datasets.

However, SSMs typically need to be *initialized* and *trained* (very) carefully. A randomly initialized SSM has suboptimal performance, but the so-called high-order polynomial projection operators (HiPPOs) (Voelker et al., 2019; Gu et al., 2020; 2023) can be used to empirically improve it. On the other hand, even a properly initialized SSM needs to be trained with care. One often needs

to set a smaller learning rate for the matrix $\mathbf{A}$ (Gu et al., 2022b), and the LTI systems require reparameterization to be trained stably (Wang & Li, 2023). To better understand these initialization and reparameterization efforts from a unified perspective, we analyze SSMs through the lens of Henkel Operator theory. Specifically, we use the Hankel singular value decomposition (HSVD) to analyze an operator defined by $\Gamma$. The decay of the Hankel singular values tells how "expressive" the LTI system is. If the Hankel singular values of $\Gamma$ decay fast, then it informally means that our $\Gamma$ cannot capture the complex patterns in the input space $\{\mathbf{u}(\cdot)\}$; in fact, the theory of reduced-order modeling (ROM) says that $\Gamma$ can be well-approximated by a reduced system with a much smaller state-space dimension $k \ll n$ (Glover, 1984).

We find that the decay of the Hankel singular values can be used for predicting the performance of an SSM, and that every previous effort in proposing a good initialization and training scheme can be viewed as an effort to avoid fast-decaying Hankel singular values. This is reminiscent of the works by Martin & Mahoney (2021) and Martin et al. (2021) that connect the singular values of the weight matrices to a deep neural network's performance. Using the Hankel singular values as heuristics, we show that an S4D model is vulnerable to losing expressiveness during training. Moreover, even with a reparameterization, an LTI system is very sensitive to a perturbation of its parameters, $\mathbf{A}$, $\mathbf{B}$, and $\mathbf{C}$, impairing the training stability of an S4D model.

Based on these insights, we propose a completely different parameterization of the LTI systems. Instead of using the continuous-time systems by matrices $\mathbf{A}$, $\mathbf{B}$ and $\mathbf{C}$, we parameterize our LTI systems by the Markov parameters of the so-called discrete **H**ankel **ope**rator (**HOPE**). A discrete Hankel operator is defined by a Hankel matrix, and is naturally associated with a discrete LTI system, and with a continuous-time system via the bilinear transform with $\Delta t = 1$ (Glover, 1984). While this continuous-time LTI system acts on our sequential data, the optimization algorithms are applied to the Markov parameters of the Hankel matrix. (See Figure 1.) We prove that unlike an LTI system parameterized by $(\mathbf{A}, \mathbf{B}, \mathbf{C}, \mathbf{D})$, one parameterized by the Markov parameters almost surely has slowly decaying singular values (see Theorem 3); moreover, it enjoys a global stability to perturbation (see Theorem 4). Hence, unlike a canonical SSM, our HOPE-SSM can be stably trained without reparameterization or reducing the learning rate, also reducing the need for hyperparameter tuning. We show

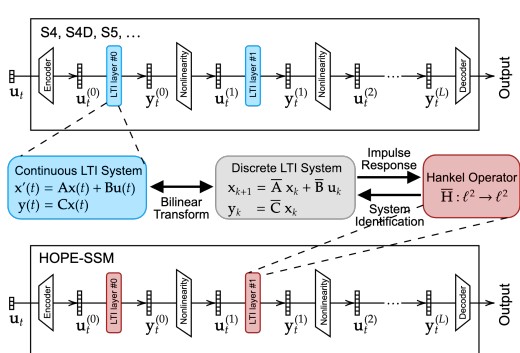

**Figure 1:** There are many equivalent ways to represent an LTI system. While most of the canonical SSMs use continuous LTI systems as their parameters, we propose to parameterize an SSM by the Markov parameters in its Hankel operator. The feedthrough matrix $\mathbf{D}$ is not shown in the diagram, but it is also a parameter of the LTI layers in both the canonical SSMs and our HOPE-SSM.

that our HOPE-SSM can be implemented by nonuniformly sampling the transfer function, enjoying the same computational complexity as the S4D model. Moreover, it requires only $1/3$ the number of parameters of an LTI system in an S4D model to parameterize that in a HOPE-SSM.

The practical benefits of our novel parameterization are improved robustness (with respect to initialization and training stability) and performance (with respect to model quality and parameter count). Moreover, we show that the memory of our HOPE-SSM does not decay (see eq. (6)) in a fixed time window, making it possible to solve tasks that involve even longer-range dependency by tuning the sampling period $\Delta t$ at discretization. This partially addresses the well-known issue that the LTI system of a canonical SSM suffers from exponentially decaying memory (Agarwal et al., 2023).

**Related Work.** Initially proposed by Voelker et al. (2019), the general idea of the HiPPO framework is to memorize the input by projecting it onto an orthogonal polynomial basis and storing the coefficients in the state vector $\mathbf{x}(t)$. This was later generalized to some different orthogonal polynomial bases in Gu et al. (2020; 2023). The S4D model uses a slightly perturbed version of the HiPPO-LegS initialization, and the effect of the perturbation was studied in Yu et al. (2024). The initialization issue was also studied in Orvieto et al. (2023) in the discrete-time setting, which provides an alternative justification of HiPPO based on the spectrum of the state matrix. While a common way to reparameterize a diagonal SSM is by training $\text{Re}(\text{diag}(\mathbf{A}))$ on a logarithmic scale (Gu et al., 2022a),

other stable reparameterizations were considered in Wang & Li (2023). A method that directly parameterizes the convolutional kernel of the discretized LTI system was presented in Fu et al. (2023). Compared to their work, in this paper, we adopt a compute-then-discretize strategy by still parameterizing the underlying continuous dynamics, making our SSM capable of handling sequences with varying lengths.

The decaying memory of RNNs and SSMs is analyzed and discussed by a wide literature (Hardt et al., 2018; Gu et al., 2020; Wang & Xue, 2024; Orvieto et al., 2024). Ways to lift the memory capacity of an SSM were considered in Wang & Li (2023) and Agarwal et al. (2023) via either reparameterizing the state matrix or applying a spectral filter. We remark that while Agarwal et al. (2023) showed that a spectral SSM is exponentially close to an SSM, our HOPE-SSM is an actual SSM containing LTI systems, and it only differs from a canonical SSM in the parameterization.

**Contributions.** Our main contributions are summarized as follows.

1. We show that high-degree LTI systems (i.e., those with slow-decaying Hankel singular values) in an SSM lead to a good performance. We justify this causal relationship using ideas in reduced-order modeling (ROM). We theoretically prove that expressive, high-degree LTI systems are scarce in the parameter space of $(\mathbf{A}, \mathbf{B}, \mathbf{C})$. Thus, they need careful designs and are numerically unstable. This explains difficulties in initializing and training SSMs.

2. We propose a new parameterization of LTI systems using the Hankel operator (HOPE), which can be implemented efficiently by nonuniformly sampling the transfer function and requires $1/3$ the number of parameters in an LTI system from an S4D model. We prove that our new parameter space is full of high-degree LTI systems. Hence, our HOPE-SSM does not suffer from the lack of expressiveness during initialization and training; moreover, it can be stably trained and has non-decaying long memory.

3. We empirically demonstrate that our HOPE-SSM is robust using the sCIFAR-10 task and that it has long-term memory using a noise-padded sCIFAR-10 dataset. We test the performance of a full-scale HOPE-SSM on the Long-Range Arena and observe that its performance exceeds that of its S4 and S4D counterparts and many other SSMs for most tasks.

## 2 PRELIMINARIES

Let $\Gamma = (\mathbf{A}, \mathbf{B}, \mathbf{C}, \mathbf{D})$ be a continuous-time LTI system defined in eq. (1). One can take a bilinear transform to obtain a discrete system $\overline{\Gamma} = (\overline{\mathbf{A}}, \overline{\mathbf{B}}, \overline{\mathbf{C}}, \overline{\mathbf{D}})$ so that the underlying dynamics is given by

$$\mathbf{x}_{k+1} = \overline{\mathbf{A}}\mathbf{x}_k + \overline{\mathbf{B}}\mathbf{u}_k, \qquad \mathbf{y}_k = \overline{\mathbf{C}}\mathbf{x}_k + \overline{\mathbf{D}}\mathbf{u}_k. \tag{2}$$

The transfer functions of $\Gamma$ and $\overline{\Gamma}$ are rational functions on the complex plane $\mathbb{C}$ defined by

$$G(s) = \mathbf{C}(s\mathbf{I} - \mathbf{A})^{-1}\mathbf{B} + \mathbf{D} \qquad \text{and} \qquad \overline{G}(z) = \overline{\mathbf{C}}(z\mathbf{I} - \overline{\mathbf{A}})^{-1}\overline{\mathbf{B}} + \overline{\mathbf{D}},$$

respectively. We usually care about the values of $G$ and $\overline{G}$ on the imaginary axis and the unit circle in $\mathbb{C}$, respectively. They "transfer" the inputs to the outputs in the frequency domain by multiplication:

$$\begin{aligned} \text{(continuous case)} \qquad & \hat{\mathbf{y}}(s) = G(is)\hat{\mathbf{u}}(s), \qquad s \in \mathbb{R}, \\ \text{(discrete case)} \qquad & \hat{\mathbf{y}}_k = \overline{G}(\boldsymbol{\omega}_k)\hat{\mathbf{u}}_k, \qquad 0 \le k \le L-1, \end{aligned} \tag{3}$$

where the hats on a function and on a vector mean the Fourier transform and the Fourier coefficients, respectively, and $\boldsymbol{\omega}_k = \exp(i2\pi k/L)$ is the $k$th Fourier node of length $L$. Throughout this paper, we assume that our LTI systems are asymptotically stable, i.e., the eigenvalues of $\mathbf{A}$ have negative real parts, or equivalently, the eigenvalues of $\overline{\mathbf{A}}$ are contained in the open unit disk in the complex plane. In this paper, we also discuss a completely different notion of stability: the numerical stability of an LTI system. This refers to the system's sensitivity to a perturbation of its parameters. We will clearly distinguish these two notions of stability by appending the adjectives "asymptotic" or "numerical".

Given a discrete LTI system $\overline{\Gamma}$, its Hankel operator is defined by a doubly infinite Hankel matrix

$$\overline{\mathbf{H}} \in \mathbb{C}^{\infty \times \infty}, \qquad \overline{\mathbf{H}} : \ell^2(\mathbb{N}) \to \ell^2(\mathbb{N}), \qquad \overline{\mathbf{H}}_{i,j} = \overline{\mathbf{C}}\,\overline{\mathbf{A}}^{i+j}\overline{\mathbf{B}}, \qquad i, j \ge 0, \tag{4}$$

where $\ell^2(\mathbb{N}) = \{(f_1, f_2, \dots) \mid f_j \in \mathbb{C}, \sum_{j=1}^{\infty} |f_j|^2 < \infty\}$. This discrete Hankel operator on $\ell^2(\mathbb{N})$ is a bounded linear operator of rank $\le n$, the number of latent states. We denote its singular

values by $\sigma_1(\overline{\mathbf{H}}) \geq \sigma_2(\overline{\mathbf{H}}) \geq \cdots \geq \sigma_n(\overline{\mathbf{H}}) \geq 0$. Analogously, one can define a continuous Hankel operator $\mathbf{H} : L^2([0, \infty)) \rightarrow L^2([0, \infty))$ given a continuous-time LTI system $\Gamma$, where $L^2([0, \infty)) = \{f : [0, \infty) \rightarrow \mathbb{C} \mid f \text{ measureable}, \int_0^\infty |f(t)|^2 dt < \infty\}$. If $\overline{\Gamma}$ is discretized from $\Gamma$ using the bilinear transform, then the singular values of $\mathbf{H}$ are equivalent to those of $\overline{\mathbf{H}}$, i.e., $\sigma_j(\mathbf{H}) = \sigma_j(\overline{\mathbf{H}})$. (See Appendix B for more details.) In this paper, we consider the decay of the Hankel singular values. To quantitatively measure how "small" a singular value is, we introduce the numerical rank of an LTI system. Given a small number $\epsilon \geq 0$, we define the $\epsilon$-rank of $\Gamma$ to be the number of relative Hankel singular values larger than $\epsilon$:

$$\text{rank}_\epsilon(\Gamma) = \max\{j \mid \sigma_j(\mathbf{H})/\sigma_1(\mathbf{H}) > \epsilon\}.$$

When $\epsilon = 0$, we reproduce the exact rank of $\mathbf{H}$, which is rarely $< n$ in the floating-point arithmetic. A small positive $\epsilon$ allows us to eliminate the small but perhaps positive singular values.

## 3 UNRAVEL A MYSTERY: HANKEL SINGULAR VALUES IN INITIALIZATION AND TRAINING

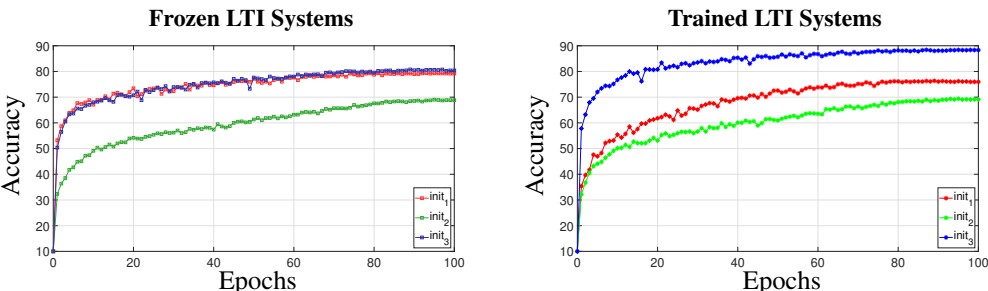

**Figure 2:** Test accuracy of the SSMs on the sCIFAR task. The LTI systems are initialized in three different ways and are either trained or untrained. We notice that when the LTI systems are initialized with $\text{init}_1$ (red), training the LTI system together with other model parameters is impairing the model accuracy. This is in contrast to SSMs initialized with $\text{init}_3$ (blue), where assigning the LTI system a small positive learning rate is helping the performance.

We begin our exploration by presenting a mystery. We use it to elicit and support the use of the Hankel singular values as protocols for explaining and predicting the success of SSMs, without stressing it as a fine-grained study of different initializations. We train an S4D model to learn the sCIFAR-10 image classification task (Krizhevsky et al., 2009; Tay et al., 2021). We consider three initialization schemes of the LTI systems in the SSM: $\text{init}_1$, $\text{init}_2$, and $\text{init}_3$. While $\text{init}_3$ is the HiPPO-LegS initialization, we treat the others as black boxes in this paper, leaving the details to Appendix D. Instead, we later explain their success or failure by measuring their Hankel singular values. For an SSM initialized using a certain $\text{init}_j$ ($1 \leq j \leq 3$), we train it in two different ways: either by freezing $\mathbf{A}, \mathbf{B}$, and $\mathbf{C}$ and only training the other model parameters or by assigning $\mathbf{A}, \mathbf{B}$, and $\mathbf{C}$ a small learning rate. The three initializations and two learning rate assignments comprise a total of six combinations, summarized in Figure 2. Besides the natural question of why we see a general difference between models initialized differently, Figure 2 raises a more intriguing mystery:

*For an SSM initialized by $\text{init}_1$, $\text{init}_2$, or $\text{init}_3$, why does training the LTI systems impair, level, or improve the performance of the model, respectively?*

To answer these questions, we study the Hankel singular values of the systems, but why are they relevant? The reason is that the Hankel singular values measure the "complexity" of an LTI system. The easiest way to see this is via ROM of the system $\Gamma$ (Adamyan et al., 1971; Glover, 1984):

*For any $k < n$, there exists a reduced-order approximation $\tilde{\Gamma} = (\tilde{\mathbf{A}}, \tilde{\mathbf{B}}, \tilde{\mathbf{C}}, \tilde{\mathbf{D}})$ with $\tilde{\mathbf{A}} \in \mathbb{C}^{k \times k}$, such that $\|G - \tilde{G}\|_\infty \leq \sum_{j=k+1}^n \sigma_j(\mathbf{H}) \leq (n-k)\sigma_{k+1}(\mathbf{H})$, where $\tilde{G}$ is the transfer function of $\tilde{\Gamma}$ and $\|\cdot\|_\infty$ is the L-infinity norm over the imaginary axis.*

In particular, if the sum of the trailing Hankel singular values $\sum_{j=k+1}^n \sigma_j(\mathbf{H})$ is small, then by eq. (3), $\Gamma$ and $\tilde{\Gamma}$ produce similar outputs on any input. Hence, ROM says that if the Hankel

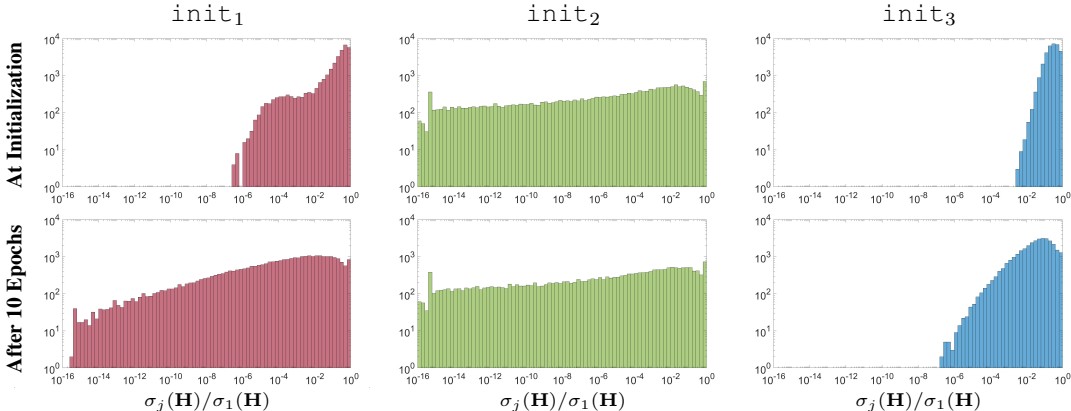

**Figure 3:** The distribution of all relative Hankel singular values $\sigma_j(\mathbf{H})/\sigma_1(\mathbf{H})$ of the LTI systems in an SSM. For each initialization, the distribution is shown both at initialization and after the SSM is trained for 10 epochs. Note that the second row only applies when the LTI systems are not frozen.

singular values decay fast, then we can replace the LTI system with a much smaller one without too much loss; in other words, most states in $\mathbf{x}(t)$ are not properly used to memorize the input.

In our experiment, each SSM has 4 layers and 128 channels. These comprise $4 \times 128 = 512$ different copies of single-input/single-output LTI systems. Every LTI has $n = 64$ latent states. These make up $512 \times 64 = 32768$ relative Hankel singular values $\sigma_j(\mathbf{H})/\sigma_1(\mathbf{H})$ to consider. In Figure 3, we show the histograms of all these relative Hankel singular values at two different stages of training. Note that the three histograms on the second row only apply when the LTI systems are trained with a small learning rate; when they are frozen, the distributions always stay the same as those at initialization. We can explain the behaviors of the three SSMs using their Hankel singular values:

1. The systems in a model initialized by $\texttt{init}_1$ initially have high numerical ranks. Hence, when the systems are untrained, they define random mappings that capture the rich content in the input data, yielding the rest of the work to other model parameters in the SSM. However, when the systems are trained, their Hankel singular values start to decay rapidly with only 27.87% of the singular values satisfying that $\sigma_j(\mathbf{H})/\sigma_1(\mathbf{H}) > 0.01$, and the systems can no longer handle a variety of distinct inputs. This makes it harder to parse complicated images in the sCIFAR-10 dataset.

2. No matter whether trained or not, the Hankel singular values of the systems in the model initialized by $\texttt{init}_2$ decay very fast, which means the systems essentially lack expressiveness and cannot capture the complicated patterns in the input space. Hence, the SSMs with both trained and untrained LTI systems could not learn the task effectively.

3. The systems in the model initialized by $\texttt{init}_3$, trained or not, have high numerical ranks. In particular, over 87.82% of the singular values satisfy that $\sigma_j(\mathbf{H})/\sigma_1(\mathbf{H}) > 0.01$ after 10 epochs. In this case, training the LTI systems allows us to accelerate the optimization.

We just established the Hankel singular values as a protocol for evaluating and predicting the performance of an SSM on tasks that involve complicated long-range dependencies. We can further derive theory to explain the patterns we observed in the histograms in Figure 3, which brings out potential weaknesses of the S4D models and motivates the development of our HOPE-SSM.

### 3.1 MANY LTI SYSTEMS HAVE LOW RANKS

We saw that LTI systems with high numerical ranks make an SSM thrive, but do we have an abundance of them? We approach this question from a random matrix theory perspective. We randomly sample an asymptotically stable LTI system. Since every diagonal entry $a_j$ of $\overline{\mathbf{A}} = \mathrm{diag}(a_1, \ldots, a_n)$ needs to be contained in the open unit disk, we cannot assume that they are sampled from a random Gaussian distribution. Instead, we assume that every $a_j$ is sampled i.i.d. from a distribution $F_a$ with

$$\mathbb{P}\left(|a_j| > (1 - \rho)\right) = \mathcal{O}(\rho^\alpha) \qquad \text{as} \qquad \rho \to 0^+$$

for some $\alpha > 0$. For example, if $a_j$ is uniformly distributed on the open unit disk, then we have $\alpha = 1$. Moreover, assume that $\overline{\mathbf{B}} \circ \overline{\mathbf{C}}^\top$ is a random vector with each entry sampled i.i.d. from a

normal distribution $\mathcal{N}(0,1)$, where $\circ$ is the Hadamard product. The skip connection matrix $\overline{\mathbf{D}}$ has no effect on the Hankel singular values. With these assumptions, we formally show that the system has a low $\epsilon$-rank with high probability.

**Theorem 1.** Given any $\epsilon > 0$, $0 < \alpha \leq 1$, and $0 < \delta \leq 1$, with probability at least $1 - \delta$, the $\epsilon$-rank of $\overline{\Gamma} = (\overline{\mathbf{A}}, \overline{\mathbf{B}}, \overline{\mathbf{C}}, \overline{\mathbf{D}})$ with $a_j \sim F_a$ i.i.d. and $b_j c_j \sim \mathcal{N}(0,1)$ i.i.d. is $\mathcal{O}(\ln(\delta^{-3/2}\epsilon^{-1}n)\,n^\beta)$, where

$$\beta \leq \frac{1}{1+\alpha} + \frac{\ln(2 + \sqrt{\ln(1/\delta)/2})}{\ln(n)}$$

and the constant in $\mathcal{O}$ is universal.

We defer the proof to Appendix E. What Theorem 1 says is that if we ignore the logarithmic factors in the $\mathcal{O}$-notation, then the $\epsilon$-rank of $\overline{\Gamma}$ scales like $n^\beta$ as $n \to \infty$, where $\beta < 1$. For example, when $a_j$ are uniformly distributed on the unit disk, we have $\beta = 1/2$ plus a small number bounded by $\ln(2 + \sqrt{\ln(1/\delta)/2})/\ln(n)$. In practice, we find that this term can almost be ignored (see Appendix H). The importance of Theorem 1 is twofold: from an expository point of view, it theoretically verifies, using our Hankel operator framework, that random initializations of LTI systems could lead to poor model performance, as observed in Gu et al. (2020) and Orvieto et al. (2023); from a practical perspective, it suggests high-rank systems are only scarce in the space of S4D model parameters. Hence, even when an LTI system is initialized with slow-decaying Hankel singular values, when $\mathbf{A}$, $\mathbf{B}$, and $\mathbf{C}$ are perturbed during training, it is at risk of losing numerical ranks and thus expressiveness. This is indeed observed in Figure 3, even for init₃.

## 3.2 LTI Systems are Numerically Unstable under Perturbations

In this section, we perform a sensitivity analysis of an S4D model, which suggests a numerical stability issue in training the model.

**Theorem 2.** Let $\Gamma = (\mathbf{A}, \mathbf{B}, \mathbf{C}, \mathbf{D})$ be a stable continuous-time LTI system, where $\mathbf{A} = \mathrm{diag}(a_1, \ldots, a_n)$ is diagonal. Let $\tilde{\Gamma} = (\tilde{\mathbf{A}}, \tilde{\mathbf{B}}, \tilde{\mathbf{C}}, \mathbf{D})$ be a perturbed stable system with $\tilde{\mathbf{A}} = \mathrm{diag}(\tilde{a}_1, \ldots, \tilde{a}_n)$. Assume there exist $\Delta_{\mathbf{A}}, \Delta_{\mathbf{B}} > 0$ such that $|a_j - \tilde{a}_j| \leq \Delta_{\mathbf{A}} \leq \min_j |\mathrm{Re}(a_j)|/2$ and $|b_j c_j - \tilde{b}_j \tilde{c}_j| \leq \min(|b_j c_j|, \Delta_{\mathbf{B}})$ for all $j = 1, \ldots, n$. Let $G$ and $\tilde{G}$ be the transfer functions of $\Gamma$ and $\tilde{\Gamma}$, respectively. Then, the following statements hold.

(a) We have

$$\|G - \tilde{G}\|_\infty \leq 4n\Delta_{\mathbf{A}} \max_j \frac{|b_j c_j|}{|\mathrm{Re}(a_j)|^2} + n\Delta_{\mathbf{B}} \max_j \frac{1}{|\mathrm{Re}(a_j)|}.$$

(b) The upper bound is tight up to a factor of $n$. That is, given *any* $\Gamma$, $\Delta_{\mathbf{A}} \leq \min_j |\mathrm{Re}(a_j)|/2$, and $\Delta_{\mathbf{B}} \leq \min_j |b_j c_j|$, there exist two systems $\tilde{\Gamma}_{\mathbf{A}}$ and $\tilde{\Gamma}_{\mathbf{B}}$, with transfer functions $\tilde{G}_{\mathbf{A}}$ and $\tilde{G}_{\mathbf{B}}$, respectively, that satisfy the above perturbation conditions and have

$$\|G - \tilde{G}_{\mathbf{A}}\|_\infty \geq \Delta_{\mathbf{A}} \max_j \frac{|b_j c_j|}{|\mathrm{Re}(a_j)|^2}, \qquad \|G - \tilde{G}_{\mathbf{B}}\|_\infty \geq \Delta_{\mathbf{B}} \max_j \frac{1}{|\mathrm{Re}(a_j)|}.$$

The proof can be found in Appendix F. Theorem 2 says that when a diagonal LTI system is trained, its numerical stability depends on two things: the proximity of its poles $a_j$ to the imaginary axis and the magnitudes of $\mathbf{B}$ and $\mathbf{C}$. We defer the discussion of $a_j$ to the end of this subsection. One may imagine that $\|G\|_\infty$ is related to $|b_j c_j|$, and hence, $|b_j c_j|$ plays no role in the *relative* numerical stability (i.e., $\|G - \tilde{G}\|_\infty / \|G\|_\infty$) of the system. However, this is not true as a system with a small $\|G\|_\infty$ may have an arbitrarily large $\mathbf{B} \circ \mathbf{C}^\top$ (see Figure 4 (Left)). Working with such a system can be numerically hazardous even at inference time due to the so-called cancellation errors (Yu & Townsend, 2024). Later, we show that our HOPE parameterization does not suffer from these issues.

Combined with the spectral information of the state matrices $\mathbf{A}$, Theorem 2 lets us explain why systems initialized by init₁ are much more sensitive to perturbation than those initialized by init₃. In Figure 4, we show the locations of the poles $a_j$ and their associated magnitudes $|b_j c_j|$. Compared to the systems from init₃, the poles of the systems from init₁ are much closer to the imaginary axis whereas the values of $|b_j c_j|$ are also much larger. Hence, by Theorem 2, they are much more sensitive to training, and therefore lose numerical ranks easily. This is indeed seen in Figure 4.

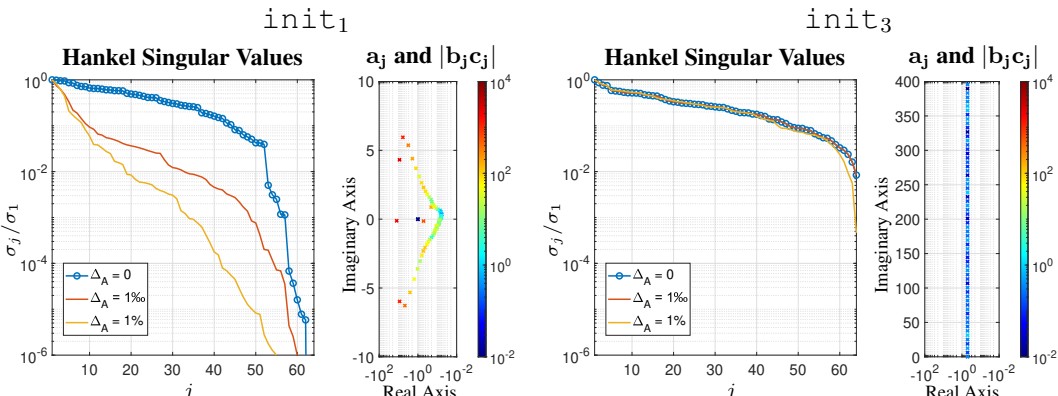

**Figure 4:** A random perturbation to the imaginary part of $\mathbf{A}$ is added to a system from `init`$_1$ and a HiPPO-LegS system from `init`$_3$. The magnitude of the perturbation is set to $0.1\%$ and $1\%$ of the original matrix $\mathbf{A}$. For each system, on the left, we show the relative Hankel singular values $\sigma_j/\sigma_1$ of the original and perturbed continuous-time systems; on the right, we plot the location of each $a_j$ in the complex plane and use the color to indicate the magnitude of its associated $|b_jc_j|$.

Theorem 2 delivers a disturbing message. As stated in Baker et al. (2015), "[the Hankel singular values] decay more rapidly the farther the $\Lambda(\mathbf{A})$ falls in the left half of the complex plane," where $\Lambda(\mathbf{A})$ is the spectrum of $\mathbf{A}$, which is equivalent to $\{a_1, \ldots, a_n\}$ in the diagonal case. Hence, many diagonal LTI systems with a high $\epsilon$-rank, i.e., those that we earlier found necessary for an SSM to capture the long-range dependency, would have eigenvalues $a_j$ close to the imaginary axis, making the system more sensitive to perturbation and thus the training less numerically stable.

## 4 HOPE-SSM: A RANKFUL, STABLE, AND LONG-MEMORY PARAMETERIZATION

Given the potential issues of an SSM discussed in section 3, we propose an entirely different parameterization of the LTI systems called HOPE. We first describe the details of our HOPE-SSM and then explain how it resolves the low-rank and numerical instability issues of an LTI system. In addition, it also benefits from long-term memory. Our strategy is to use a Hankel matrix defined in eq. (4) to parameterize an LTI system. Instead of having $\mathbf{A}$, $\mathbf{B}$, $\mathbf{C}$, $\mathbf{D}$, and $\Delta t$ as the model parameters, we now have a vector $\mathbf{h}$ of length $n$, the skip connection $\mathbf{D}$, and $\Delta t$ as our model parameters. Hence, we use $n$ complex parameters from $\mathbf{h}$ to replace the $3n$ (resp. $4n$) complex parameters from $\mathbf{A}$, $\mathbf{B}$, and $\mathbf{C}$ in S4D (resp. S4); [1] We remark that, as mentioned in the introduction, our model only modifies the LTI systems in an SSM. That is, it requires $1/3$ the number of parameters in an S4D model *for each LTI system*. Since there are other components in an SSM (e.g., the encoder and the decoder), we do not compress the number of parameters in the entire model by a factor of $1/3$. The finite Hankel matrix $\overline{\mathbf{H}} \in \mathbb{C}^{n \times n}$ is then defined by the Markov parameters in $\mathbf{h}$:

$$\overline{\mathbf{H}}_{i,j} = \mathbb{1}_{\{i+j<n\}}\mathbf{h}_{i+j}. \tag{5}$$

In an S4D model, we start with $\mathbf{A}$, $\mathbf{B}$, and $\mathbf{C}$ so that $\mathbf{h}_{i+j} = \overline{\mathbf{C}}\,\overline{\mathbf{A}}^{i+j}\overline{\mathbf{B}}$; in a HOPE-SSM, we start with $\mathbf{h}_{i+j}$ and the matrix $\overline{\mathbf{H}}$ corresponds to a discrete LTI system $\overline{\Gamma}$, which is further associated with a continuous-time system $\Gamma$ via the bilinear transform with $\Delta t = 1$. Notably, in our HOPE-SSM, it is the continuous system $\Gamma$ that takes the role of $(\mathbf{A}, \mathbf{B}, \mathbf{C})$ in a canonical SSM, e.g., S4 and S4D. In particular, $\Gamma$ is then discretized with a trainable sampling period $\Delta t$ for a discrete sequential input. Algorithm 1 computes HOPE, and we leave a detailed derivation in Appendix I. We also present a flowchart in Figure 5 to better illustrate the concept. We emphasize that compared to most convolutional-type models (Li et al., 2023), which parameterize a discrete convolutional kernel, our method parameterizes a continuous convolutional kernel. Given the discrete input sequence, we then discretize this convolutional kernel with respect to a trainable $\Delta t$, making the model adaptable to sequences of varying lengths (see Table 3 for an ablation study).

---

[1]There are different variants of S4D models that combine $\mathbf{B}$ and $\mathbf{C}$ into a single complex vector or use fewer copies of $\mathbf{A}$ than the number of channels. Here, we compare HOPE to a vanilla S4D model.

---

**Algorithm 1** Computing the output of an LTI system parameterized by its Hankel matrix.

---

**Input:** an input sequence $\mathbf{u} \in \mathbb{R}^L$, the Markov parameters of a Hankel matrix $\mathbf{h} \in \mathbb{C}^n$, and a sampling period $\Delta t > 0$.
**Output:** the output $\mathbf{y} \in \mathbb{R}^L$ of the LTI system defined by $\mathbf{h}$ given input $\mathbf{u}$ and sampling period $\Delta t$.

1: $\boldsymbol{\omega} \leftarrow \exp\left(2\pi i \frac{0:(L-1)}{L}\right)$                                      {create FFT nodes}
2: $\mathbf{s} \leftarrow (\boldsymbol{\omega} - 1)./(\boldsymbol{\omega} + 1)$           {convert to the $s$-domain, where ./ is the entrywise division}
3: $\mathbf{s} \leftarrow \mathbf{s}/\Delta t$                            {scale the frequency domain in the $s$-plane}
4: $\boldsymbol{\omega} \leftarrow (1 + \mathbf{s})./(1 - \mathbf{s})$                          {convert back to the $z$-plane}
5: $\mathbf{g} \leftarrow \texttt{zeros}(L)$                       {store samples of the transfer function}
6: **for** $i = 0 : (n-1)$ **do**
7:    $\mathbf{g} \leftarrow \mathbf{g} + \mathbf{h}_i \cdot (\boldsymbol{\omega}.^{\wedge}(-i-1))$    {compute the $i$th moment, where $.^{\wedge}$ is the entrywise power}
8: **end for**
9: $\mathbf{y} \leftarrow \text{Re}\left(\texttt{iFFT}(\texttt{FFT}(\mathbf{u}) \circ \mathbf{g})\right)$              {$\circ$ is the entrywise (i.e., Hadamard) product}

---

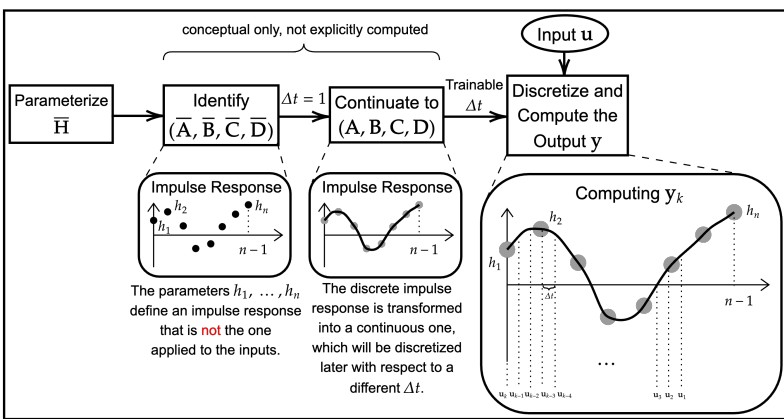

**Figure 5:** A flowchart that illustrates how the Hankel matrix $\overline{\mathbf{H}}$ is used to compute an output sequence $\mathbf{y}$ given an input sequence $\mathbf{u}$.

With $L$ processors, we can compute the entries of $\mathbf{g}$ in Algorithm 1 in parallel, each of which takes $\mathcal{O}(n)$ time. Computing the FFT and iFFT takes $\mathcal{O}(L \log L)$. Overall, the evaluation of the Hankel-parameterized LTI system takes $\tilde{\mathcal{O}}(L + n)$ time and $\mathcal{O}(L)$ space, which agree with the complexities of the S4 and S4D models. To make the model recurrent during the inference time, one can either identify a system $(\mathbf{A}, \mathbf{B}, \mathbf{C})$ whose Hankel operator is $\mathbf{H}$ and compute as if it is an S4D model (Kramer & Gorodetsky, 2018; Aumann & Gosea, 2023), or directly compute the convolutional kernel using the iFFT of $\mathbf{g}$ in Algorithm 1.

**Advantage I: A HOPE-SSM has a High Numerical Rank.** We can build upon Bryc et al. (2006) and Nekrutkin (2013) to prove the following result about random Hankel matrices.

**Theorem 3.** Let $\mathbf{h}_1, \mathbf{h}_2, \ldots$ be a sequence of i.i.d. random variables with mean 0 and variance 1 that have finite third moments. We almost surely have that for any $\epsilon > 0$, the $(\epsilon/\sqrt{\ln(n)})$-rank of $\overline{\mathbf{H}}_n$ is $\Omega(n)$, where $\overline{\mathbf{H}}_n$ is the $n \times n$ Hankel matrix defined in eq. (5).

Hence, if we ignore the logarithmic factor of $\sqrt{\ln(n)}$, then the numerical rank of a random Hankel matrix should be proportional to $n$ as $n \to \infty$. That means unlike the S4D model, we can always randomly initialize a high-rank LTI system with HOPE; even better, a system parameterized by HOPE does not lose rank during training (see Figure 6), because the low-rank systems are themselves rare in the space of the Markov parameters $\mathbf{h}$ (but not in the space of $(\mathbf{A}, \mathbf{B}, \mathbf{C})$).

**Advantage II: A HOPE-SSM is Numerically Stable under Perturbation.** Unlike the LTI system, a Hankel matrix $\overline{\mathbf{H}}$ is very numerically stable under perturbation, as shown in the following theorem.

**Theorem 4.** Let $\mathbf{h} \in \mathbb{C}^{n \times 1}$ be a vector and $G$ be the transfer function defined in eq. (11). Suppose we perturb $\mathbf{h}$ to $\tilde{\mathbf{h}}$ and let $\tilde{G}$ be its transfer function. Then, we have $\|G - \tilde{G}\|_\infty \leq \sqrt{n}\|\mathbf{h} - \tilde{\mathbf{h}}\|_2$.

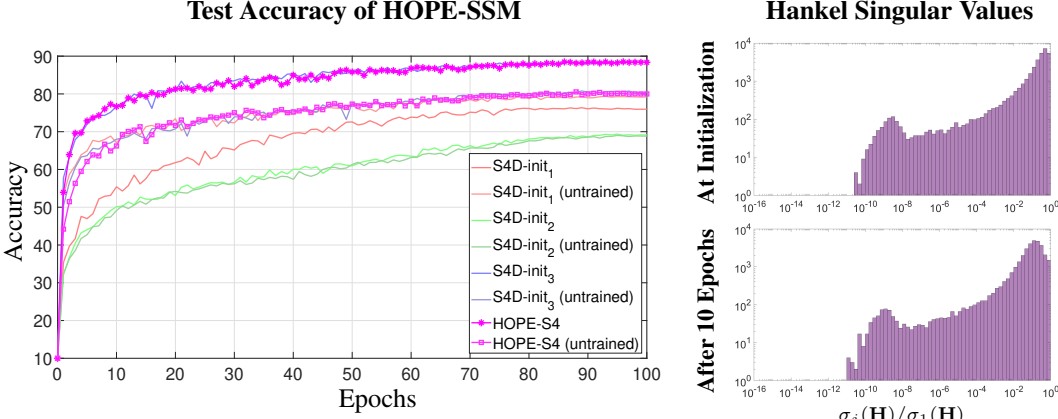

**Figure 6:** The test accuracy of the HOPE-SSM on the sCIFAR-10 task and the evolution of the Hankel singular values of the model. The plots are to be compared with Figure 2 and Figure 3.

Therefore, the numerical stability of the HOPE-SSM does not depend on the parameter $\mathbf{h}$ itself. Consequently, our HOPE-SSMs are trained without having to rescale the Markov parameters $\mathbf{h}$

**Advantage III: A HOPE-SSM's Memory Does Not Fade.** While many LTI systems are tailored for long-term memory (Voelker et al., 2019; Gu et al., 2020), an asymptotically stable system must inevitably suffer from an exponential decay in its memory (Agarwal et al., 2023). This can be manifested by the Hankel matrix: since $\begin{bmatrix} \mathbf{y}_0^\top & \mathbf{y}_1^\top & \cdots \end{bmatrix}^\top = \overline{\mathbf{H}} \begin{bmatrix} \mathbf{u}_{-1}^\top & \mathbf{u}_{-2}^\top & \cdots \end{bmatrix}^\top$, we can write

$$\mathbf{y}_j = \sum_{k=1}^{\infty} \overline{\mathbf{H}}_{j,k-1}\mathbf{u}_{-k} = \sum_{k=1}^{\infty} \overline{\mathbf{H}}_{0,k+j-1}\mathbf{u}_{-k}, \qquad j \geq 0. \tag{6}$$

That means the effect of the input $\mathbf{u}_{-k}$ on the output $\mathbf{y}_j$ depends on the magnitude of $\overline{\mathbf{H}}_{0,k+j-1}$. Hence, the decay of $|\overline{\mathbf{H}}_{0,t}|$ gives us a measurement of the decay of the memory. For a discrete LTI system, we have by definition that $\overline{\mathbf{H}}_{0,t} = \overline{\mathbf{C}}\,\overline{\mathbf{A}}^t\overline{\mathbf{B}}$. Since the spectrum of $\overline{\mathbf{A}}$ for an S4D model, regardless of $\Delta t$, is contained in the open unit disk, $\overline{\mathbf{A}}$ is a contraction operator and the "memory" of the system decays as time goes by. Asymptotically in time, this decay of $|\overline{\mathbf{H}}_{0,t}|$ is exponential, and for most systems, it starts at the very beginning when $t$ increases from $0$.[2] However, if we parameterize the LTI system by the Markov parameters in the Hankel matrix, then $|\overline{\mathbf{H}}_{0,t}| = \mathbf{h}_t$, which does not have to decay as long as $t < n$. We acknowledge that when $t \geq n$, we have $\overline{\mathbf{H}}_{0,t} = 0$, which means the LTI system has essentially no memory after time $n$. We remark, however, that our HOPE-SSM utilizes the continuous-time LTI system associated with a Hankel matrix $\overline{\mathbf{H}}$. That is, from a continuous-time perspective, unlike S4D, our system's memory does not decay for $t \in [0, n]$ (see Figure 7). Hence, even with a long sequence whose length is much larger than $n$, by setting the sampling period $\Delta t$ to be small enough, our HOPE-SSM could still enjoy non-decaying memory.

## 5 EXPERIMENTS AND DISCUSSIONS

**Experiment I: Singular Values of a HOPE-SSM.** In this section, we implement a randomly initialized HOPE-SSM to learn the sCIFAR-10 task. We use the same model hyperparameters as the S4D models in section 3. In particular, the Hankel matrices in this model are 64-by-64. We randomly initialize the Hankel matrix and do not set a smaller learning rate for the Hankel matrix entries $\mathbf{h}$, i.e., all model parameters except for $\Delta t$ have the same learning rate. In Figure 6, we show the test accuracy and the Hankel singular values of the HOPE-SSM. Compared to Figure 2, a random HOPE-SSM can be trained to a high accuracy. In addition, training does not reduce the numerical rank of a Hankel matrix. This corroborates our findings in Theorem 3 and Theorem 4. Note that since we used the log-log scale to plot the histograms, the small relative Hankel singular values make up a misleadingly large portion of densities in Figure 6 than they really do.

---

[2]In particular, this is the case for HiPPO-LegS (see Figure 7).

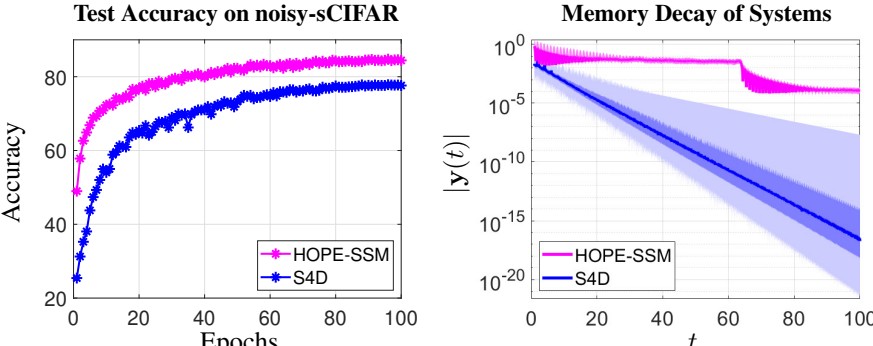

**Figure 7:** Left: the test accuracies of the S4D model and our HOPE-SSM on the noisy-sCIFAR task. Right: a unit impulse at $t = 0$ is acted on the LTI system. The plot shows the decay of $|\mathbf{y}(t)|$ as $t$ increases. Data are collected for all 512 LTI systems in a trained model. The dark curve shows the median of $|\mathbf{y}(t)|$ over the 512 numbers. The darkly shaded region is from the first quartile to the third quartile. The lightly shaded region is from the minimum to the maximum.

**Experiment II: HOPE-SSMs Have Long Memory.** In section 4, we claim that a HOPE-SSM benefits from non-decaying memory. We show this experimentally in this section. To do so, for each flattened picture in the sCIFAR-10 dataset, which contains 1024 vectors of length 3, we append a random sequence of 1024 vectors of length 3 to the end of it. The goal is still to classify an image by its first 1024 pixels. We call this task noise-padded sCIFAR-10. This task obviously requires a long memory so that the earlier data can still be retrieved after the noises are taken. We train both an S4D model and our HOPE-SSM to learn this task, using a common sampling period of $\Delta t = 0.1$ (see Appendix K.2 for different values of $\Delta t$). We see in Figure 7 that the HOPE-SSM significantly outperforms the S4D model on this task. To understand this gap, we give a unit impulse at $t = 0$ to the LTI systems in both the trained S4D model and our HOPE-SSM. We watch how the impulse response $\mathbf{y}(t)$ decays as time goes by. In Figure 7, we see that the memory of the trained S4D model decays exponentially while that of the trained HOPE-SSM does not decay at all for $t \in [0, 64]$, where $n = 64$ is the size of the Hankel matrix. This is exactly what we expected in eq. (6).

**Experiment III: Performance in the Long-Range Arena.** Finally, we present the performance of HOPE-SSM on large datasets. We use the same model architecture as in the S4D paper (Gu et al., 2022a), except that we replace the LTI blocks with our HOPE blocks. The specific model and training hyperparameters are reported in Table 2 in Appendix K. We show the performance of our model in Table 1. We see that our HOPE-SSM outperforms most sequential models on many tasks.

**Table 1:** Test accuracies in the Long-Range Arena of our HOPE-SSM and other models. We report the median and the standard deviation of executions with 5 random seeds. The bold (resp. underlined) numbers indicate the best (resp. second best) performance on a task. An entry is left blank if no result is found. We use the same model sizes as those in the S4D model.

| Model | ListOps | Text | Retrieval | Image | Pathfinder | Path-X | Avg. |
|---|---|---|---|---|---|---|---|
| DSS (Gupta et al., 2022) | 57.60 | 76.60 | 87.60 | 85.80 | 84.10 | 85.00 | 79.45 |
| S4++ (Qi et al., 2024) | 57.30 | 86.28 | 84.82 | 82.91 | 80.24 | - | - |
| Reg. S4D (Liu & Li, 2024) | 61.48 | 88.19 | 91.25 | 88.12 | 94.93 | 95.63 | 86.60 |
| Spectral SSM (Agarwal et al., 2023) | 60.33 | 89.60 | 90.00 | - | 95.60 | 90.10 | - |
| Liquid S4 (Hasani et al., 2023) | **62.75** | 89.02 | 91.20 | **89.50** | 94.80 | 96.66 | 87.32 |
| S5 (Smith et al., 2023) | 62.15 | 89.31 | 91.40 | 88.00 | 95.33 | **98.58** | 87.46 |
| S4 (Gu et al., 2022b) | 59.60 | 86.82 | 90.90 | 88.65 | 94.20 | 96.35 | 86.09 |
| S4D (Gu et al., 2022a) | 60.47 | 86.18 | 89.46 | 88.19 | 93.06 | 91.95 | 84.89 |
| HOPE-SSM | 62.60
±0.92 | **89.83**
±0.37 | **91.80**
±0.11 | 88.68
±0.44 | **95.73**
±0.28 | 98.45
±0.17 | **87.85** |

## 6 CONCLUSION

In this paper, we presented a new theory based on the Hankel singular values to understand the difficulties in initializing and training an SSM. We proposed a new parameterization scheme, called HOPE, that is based on the Markov parameters in a discrete Hankel operator. We proved that a HOPE-SSM can be robustly initialized and trained and has a long memory.

ACKNOWLEDGMENTS

AY was supported by the SciAI Center, funded by the Office of Naval Research under Grant Number N00014-23-1-2729. NBE would like to acknowledge NSF, under Grant No. 2319621, and the U.S. Department of Energy, under Contract Number DE-AC02-05CH11231 and DE-AC02-05CH11231, for providing partial support of this work. We thank the anonymous reviewers for their comments that helped improve this paper. We would also like to thank Alex Townsend, Anil Damle, and Sarah Dean for some inspiring discussions.

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

## A  More background on the LTI system and Hankel operators

The purpose of this section is to provide a more thorough exposition of section 2, which leaves out some concepts that are unnecessary in order to understand the main text. Let $\Gamma = (\mathbf{A}, \mathbf{B}, \mathbf{C}, \mathbf{D})$ be a continuous-time LTI system defined in eq. (1). One can take a bilinear transformation to obtain a discrete LTI system:

$$\overline{\mathbf{A}} = (\mathbf{I}+\mathbf{A})(\mathbf{I}-\mathbf{A})^{-1}, \quad \overline{\mathbf{B}} = (\mathbf{I}+\overline{\mathbf{A}})\mathbf{B}/\sqrt{2}, \quad \overline{\mathbf{C}} = \mathbf{C}(\mathbf{I}+\overline{\mathbf{A}})/\sqrt{2}, \quad \overline{\mathbf{D}} = \mathbf{D}+\overline{\mathbf{C}}(\mathbf{I}-\overline{\mathbf{A}})^{-1}\overline{\mathbf{B}}.$$

The bilinear transformation is invertible and defines a one-to-one correspondence between $\Gamma$ in eq. (1) and $\overline{\Gamma} = (\overline{\mathbf{A}}, \overline{\mathbf{B}}, \overline{\mathbf{C}}, \overline{\mathbf{D}})$ given by

$$\overline{\mathbf{x}}_{k+1} = \overline{\mathbf{A}}\overline{\mathbf{x}}_k + \overline{\mathbf{B}}\overline{\mathbf{u}}_k,$$
$$\overline{\mathbf{y}}_k = \overline{\mathbf{C}}\overline{\mathbf{x}}_k + \overline{\mathbf{D}}\overline{\mathbf{u}}_k.$$

The transfer functions of $\Gamma$ and $\overline{\Gamma}$ are

$$G(s) = \mathbf{C}(s\mathbf{I} - \mathbf{A})^{-1}\mathbf{B} + \mathbf{D} \qquad \text{and} \qquad \overline{G}(z) = \overline{\mathbf{C}}(z\mathbf{I} - \overline{\mathbf{A}})^{-1}\overline{\mathbf{B}} + \overline{\mathbf{D}},$$

respectively. The two transfer functions are equivalent via a Mobius transformation:

$$G(s) = \overline{G}\left((1+s)/(1-s)\right) \qquad \Leftrightarrow \qquad \overline{G}(z) = G\left((z-1)/(z+1)\right).$$

The importance of the transfer functions is that they bring the inputs to the outputs in the Laplace domain by multiplication, which reduces to the Fourier domain if we assume that $\mathbf{u}$ is bounded and compactly supported:

$$\hat{\mathbf{y}}(s) = G(is)\hat{\mathbf{u}}(s), \qquad \hat{\overline{\mathbf{y}}} = \overline{G}(\boldsymbol{\omega})\hat{\overline{\mathbf{u}}}, \tag{7}$$

where the hat of a function and a vector means the Fourier transform and the Fourier coefficients, respectively, and $\boldsymbol{\omega}$ is the vector of roots of unity.

Given a continuous-time LTI system $\Gamma$, one can define its Hankel operator by

$$\mathbf{H} : L^2(0, \infty) \to L^2(0, \infty), \qquad (\mathbf{H}\mathbf{v})(t) = \int_0^\infty \mathbf{C}\exp((t+\tau)\mathbf{A})\mathbf{B}\mathbf{v}(\tau)d\tau.$$

The Hankel operator maps the past inputs to the future outputs, i.e., if $\mathbf{u}(t) = 0$ for $t \geq 0$ and we define $\mathbf{v}(t) = \mathbf{u}(-t)$, then we have $\mathbf{y}(t) = (\mathbf{H}\mathbf{v})(t)$. Analogously, the Hankel matrix of a discrete LTI system $\overline{\overline{\Gamma}}$ is a doubly infinite matrix defined by

$$\overline{\mathbf{H}} : \ell^2 \to \ell^2, \qquad \overline{\mathbf{H}}_{i,j} = \overline{\mathbf{C}}\,\overline{\mathbf{A}}^{i+j}\overline{\mathbf{B}}, \qquad i, j \geq 0.$$

The Hankel matrix has the similar physical interpretation: if $\overline{\mathbf{u}}_k = 0$ for all $k \geq 0$, then we have $\begin{bmatrix}\overline{\mathbf{y}}_0^\top & \overline{\mathbf{y}}_1^\top & \cdots\end{bmatrix}^\top = \overline{\mathbf{H}}\begin{bmatrix}\overline{\mathbf{u}}_{-1}^\top & \overline{\mathbf{u}}_{-2}^\top & \cdots\end{bmatrix}^\top$. Both $\mathbf{H}$ and $\overline{\mathbf{H}}$ are bounded linear operators of rank $\leq n$, the number of latent states. In fact, the singular values $\sigma_1(\mathbf{H}) \geq \sigma_2(\mathbf{H}) \geq \cdots \geq \sigma_n(\mathbf{H}) \geq 0$ of $\mathbf{H}$ are equivalent to those of $\overline{\mathbf{H}}$, and they are called the Hankel singular values of $\Gamma$ and $\overline{\Gamma}$.

## B  More background on Hankel singular values

In section 2, we define the Hankel singular values $\sigma_1, \ldots, \sigma_n$ to be the singular values of the finite-rank bounded linear Hankel operator on a separable Hilbert space. Throughout the paper, we treated them as black boxes and used their distributions to understand the performance of SSMs. The goal of this section is to open the black boxes and introduce more useful properties of the Hankel singular values. Note that the proof of Theorem 1 to be presented in Appendix E heavily relies on the background discussed in this section. The concepts in this section can be presented on a continuous-time LTI system $(\mathbf{A}, \mathbf{B}, \mathbf{C}, \mathbf{D})$ or analogously on a discrete LTI system $(\overline{\mathbf{A}}, \overline{\mathbf{B}}, \overline{\mathbf{C}}, \overline{\mathbf{D}})$. We focus mainly on a continuous-time system for cleanliness. Since $\mathbf{D}$ has no effect on the Hankel singular values, we further assume that $\mathbf{D} = \mathbf{0}$ and write only $\Gamma = (\mathbf{A}, \mathbf{B}, \mathbf{C})$.

## B.1 HANKEL SINGULAR VALUES FROM BALANCED REALIZATION

One of the most popular ways to compute the Hankel singular values is via the so-called balanced realization. Assume $(\mathbf{A}, \mathbf{B}, \mathbf{C})$ is asymptotically stable, i.e. the spectrum of $\mathbf{A}$ is contained in the open left half-plane. Then, there exist two Hermitian and positive semi-definite matrices $\mathbf{P} \in \mathbb{C}^{n \times n}$ and $\mathbf{Q} \in \mathbb{C}^{n \times n}$ such that

$$
\begin{aligned}
\mathbf{AP} + \mathbf{PA}^* + \mathbf{BB}^* &= \mathbf{0}, \\
\mathbf{A}^*\mathbf{Q} + \mathbf{QA} + \mathbf{C}^*\mathbf{C} &= \mathbf{0}.
\end{aligned}
\tag{8}
$$

The equations in eq. (8) are called the Lyapunov equations and $\mathbf{P}$ and $\mathbf{Q}$ are called the controllability Gramian and the observability Gramian, respectively. They can be explicitly expressed by the following matrix integrals:

$$
\mathbf{P} = \int_0^\infty \exp(\mathbf{A}t)\mathbf{BB}^*\exp(\mathbf{A}^*t)dt, \qquad \mathbf{Q} = \int_0^\infty \exp(\mathbf{A}^*t)\mathbf{C}^*\mathbf{C}\exp(\mathbf{A}t)dt.
\tag{9}
$$

The controllability Gramian $\mathbf{P}$ is positive definite if and only if the system is controllable, i.e., for any $\mathbf{x}_0, \mathbf{x}_1 \in \mathbb{C}^n$ and any $T > 0$, there exists an input $\mathbf{u}$ on $[0, T]$ that makes $\mathbf{x}(T) = \mathbf{x}_1$ when $\mathbf{x}(0) = \mathbf{x}_0$. Likewise, $\mathbf{Q}$ is positive definite if and only if the system is observable, i.e., for any $T > 0$, the initial state $\mathbf{x}(0)$ can be determined by the input $\mathbf{u}$ and the output $\mathbf{y}$ on $[0, T]$ (Zhou & Doyle, 1998).

The singular values of $\mathbf{PQ}$ turn out to be exactly the squares of the Hankel singular values, i.e., $\sigma_1^2, \ldots, \sigma_n^2$, of the system $\Gamma$. In general, $\mathbf{P}$ and $\mathbf{Q}$ are dense matrices. However, one can use the so-called balanced realization algorithm (Laub et al., 1987) to compute an equivalent system $\Gamma_b = (\mathbf{A}_b, \mathbf{B}_b, \mathbf{C}_b) = (\mathbf{V}^{-1}\mathbf{AV}, \mathbf{V}^{-1}\mathbf{B}, \mathbf{CV})$, where $\mathbf{V} \in \mathbb{C}^{n \times n}$ is an invertible matrix, so that its Gramians $\mathbf{P}_b$ and $\mathbf{Q}_b$ are equivalent and diagonal, i.e.,

$$
\mathbf{P}_b = \mathbf{Q}_b = \mathrm{diag}(\sigma_1, \ldots, \sigma_n).
$$

Balanced realization is an algebraic method that is based on singular value decompositions (SVDs). In practice, it is a very popular method to compute the Hankel singular values of a system.

## B.2 HANKEL SINGULAR VALUES AS A RATIONAL APPROXIMATION PROBLEM

The balanced realization gives us a good way to compute the Hankel singular values in practice. However, it does not offer too much insight in theoretically analyzing them. The theory of Hankel singular values is usually derived via function approximation. Here, we introduce how a Hankel singular value can be reframed as the solution to a rational approximation problem. To this end, we let $H_+^\infty$ (resp. $H_-^\infty$) be the Hardy space with functions $h : \mathbb{C} \to \mathbb{C}^{m \times p}$ that are bounded and analytic in the open right (resp. left) half-plane. The non-tangential limit of $h$ exists almost everywhere on the imaginary axis, and by the Maximum Modulus Principle, we must have

$$
\|h\|_\infty := \mathrm{ess}\sup_{\mathrm{Re}(s)=0}\|h(s)\|_2 = \sup_{\mathrm{Re}(s)>0}\|h(s)\|_2, \qquad h \in H_+^\infty
$$

and

$$
\|h\|_\infty := \mathrm{ess}\sup_{\mathrm{Re}(s)=0}\|h(s)\|_2 = \sup_{\mathrm{Re}(s)<0}\|h(s)\|_2, \qquad h \in H_-^\infty.
$$

The Adamyan–Arov–Krein (AAK) Theory (Adamyan et al., 1971) says the following:

$$
\sigma_{k+1} = \inf_{R_k, F}\|G - R_k - F\|_\infty, \qquad 0 \le k \le n-1,
\tag{10}
$$

where $R_k$ ranges over all rational functions with at most $k$ poles, all contained in the open left half-plane, and $F$ ranges over $H_-^\infty$. In fact, as discussed in section 2, $R_k$ is the transfer function of a reduced system $\tilde{\Gamma} = (\tilde{\mathbf{A}}, \tilde{\mathbf{B}}, \tilde{\mathbf{C}}, \tilde{\mathbf{D}})$ with $\tilde{\mathbf{A}} \in \mathbb{C}^{k \times k}$. Hence, eq. (10) tells us that if the transfer function of $\Gamma$ can be well-approximated by rational functions, then it has fast-decaying Hankel singular values. The other direction is not true, due to the existence of $F$. That is, if the Hankel singular values decay fast, then it does not necessarily mean that $G$ can be well-approximated by rational functions. However, section 2 shows that this is true if the sum of the tails of the Hankel singular values decay rapidly.

## C    COMPARISON TO RELATED WORKS

One of the advantages of our HOPE-SSM is its long-memory capacity (see section 4). Notably, there have been two prior works that also consider the memory issues of an SSM (Wang & Li, 2023; Agarwal et al., 2023). In this section, we briefly introduce the two works and compare them to our HOPE-SSM. The Stable-SSM proposed by Wang & Li (2023) is based on a theoretical analysis of stable representations of long-memory SSMs. In that work, it is shown that without any reparameterization of the matrix $\mathbf{A}$, an LTI system cannot stably represent dynamics with long memory. That is, one can slightly perturb the LTI system to break the long memory. Based on the theory, it then proposes, in addition to the traditional $\log(\mathrm{Re}(\mathbf{A}))$ parameterization of the real parts of $\mathbf{A}$, different parameterizations to enhance of stability of long-memory representations. Our HOPE-SSMs are also based on different parameterizations of LTI systems. While a Stable-SSM reparameterizes $\mathbf{A}$, $\mathbf{B}$, and $\mathbf{C}$, we totally give up representing the system matrices but instead rely on the Hankel operator. That is, a HOPE-SSM has in addition two advantages (I and II) shown in section 4. Moreover, it is easy to directly change the memory decay of a HOPE-SSM to account for recency bias (see Appendix L).

The Spectral-SSM proposed in Agarwal et al. (2023) is based on the spectral filtering mechanism. The idea is to project the input sequence adaptively onto a predefined basis and then compute the output by taking a linear combination of the projections. The model is convolution-based, similar to HOPE. However, the two models attack the varying-length sequences in different ways. A Spectral-SSM handles varying-length sequences by fixing a truncated set of basis that is independent of the input length, while a HOPE-SSM defines a continuous-time convolution kernel to be discretized by trainable a $\Delta t$ to handle sequences of different lengths. Finally, a Spectral-SSM may not be equivalent to a standard SSM with LTI systems, whereas a HOPE-SSM is indeed one with LTI systems.

## D    THREE DIFFERENT INITIALIZATION SCHEMES

In this section, we explain the details of $\mathrm{init}_1$, $\mathrm{init}_2$, and $\mathrm{init}_3$ in the main text. We stress again that we do not claim that these initialization schemes are representative. Indeed, we only use them to elicit the story of the Hankel singular values. As mentioned in the main text, $\mathrm{init}_3$ is the HiPPO-LegS initialization scheme (Gu et al., 2020). To sample a random with $\mathrm{init}_1$, we create random samples of the transfer function $\{(is_j, G_j)\}_{j=1}^{N}$, where $s_j \in \mathbb{R}$ and $G_j \in \mathbb{C}$. We then identify a system $\Gamma$ whose transfer function $G$ satisfies that $G(is_j) \approx G_j$. The way we identify this model is via the so-called AAA algorithm (Nakatsukasa et al., 2018; Aumann & Gosea, 2023). For $\mathrm{init}_2$, we sample a discrete system by assuming the diagonal entries of $\overline{\mathbf{A}} = \mathrm{diag}(a_1, \ldots, a_n)$ are uniformly sampled on the unit disk and $\overline{\mathbf{B}} \circ \overline{\mathbf{C}}^{\top}$ is a random vector with each entry sampled i.i.d. from a normal distribution $\mathcal{N}(0, 1)$. We then compute the corresponding continuous-time system from the discrete system using the bilinear transform.

## E    PROOF OF THEOREM 1

Let $\overline{G}$ be the transfer function of a random system $\overline{\Gamma}_2$, i.e.,

$$\overline{G}(z) = \overline{\mathbf{C}}(z\mathbf{I} - \overline{\mathbf{A}})^{-1}\overline{\mathbf{B}} + \overline{\mathbf{D}}.$$

By the AAK theory, the Hankel singular values of $\overline{\Gamma}$ can be studied via the rational approximation of $\overline{G}$. Since the matrix $\overline{\mathbf{D}}$ does not affect the Hankel singular values of a system, we assume, without loss of generality, that $\overline{\mathbf{D}} = \mathbf{0}$. We let $\sigma_1, \ldots, \sigma_n$ be the random variables that are equal to the singular values of the random LTI system $\overline{\Gamma}_2$. We approach Theorem 1 in three steps:

1. We separate out the poles of the transfer function $G$ of $(\overline{\mathbf{A}}, \overline{\mathbf{B}}, \overline{\mathbf{C}}, \mathbf{0})$ that are close to the boundary of the unit disk. This breaks $\overline{G}$ into two low-rank systems $\overline{G}_1$ and $\overline{G}_2$, where $\overline{G}_1$ is low-rank because it has few poles and $\overline{G}_2$ is low-rank because its poles are far away from the boundary.

2. We then estimate the decay of the singular values of $\overline{G}_2$ using the information of the maximum moduli of its poles.

3. Finally, we control $\sigma_1$ in probability. This gives a control on the relative singular values.

We will make these three steps into three lemmas and use them to derive the result at the end.

**Lemma 1.** Given $\gamma > 0$, with probability at least $1 - \delta$, there are at most $n^\beta$ poles of $\overline{G}(z)$ outside the disk $D(0, 1 - n^{-\gamma})$, where

$$\beta = 1 + \log_n\left(n^{-\gamma\alpha} + \sqrt{\frac{\ln(1/\delta)}{2n}}\right).$$

*Proof.* Let $Z$ be the number of poles inside $D(0, 1 - n^{-\gamma})$. Then, $Z$ has a binomial distribution

$$Z \sim B(n, 1 - n^{-\gamma\alpha}).$$

From Hoeffding's inequality, we have

$$\mathbb{P}(Z \leq n - n^\beta) \leq \exp\left(-2n\left(1 - n^{-\gamma\alpha} - \frac{n - n^\beta}{n}\right)^2\right) = \exp\left(-2n\left(-n^{-\gamma\alpha} + n^{\beta-1}\right)^2\right)$$

Set

$$\beta = 1 + \log_n\left(n^{-\gamma\alpha} + \sqrt{\frac{\ln(1/\delta)}{2n}}\right) = 1 - \gamma\alpha + \log_n\left(1 + \sqrt{\frac{\ln(1/\delta)}{2n^{1-2\gamma\alpha}}}\right).$$

Then, we have

$$n^{\beta-1} = n^{-\gamma\alpha} + \sqrt{\frac{\ln(1/\delta)}{2n}}$$

so that

$$\mathbb{P}(Z \leq n - n^\beta) \leq \exp\left(-2n\frac{\ln(1/\delta)}{2n}\right) = \delta.$$

This finishes the proof. $\qquad\square$

**Lemma 2.** For any $\gamma > 0$, let the random variable $k$ be the number of poles of $\overline{G}(z)$ inside $D(0, 1 - n^{-\gamma})$. Let $\kappa > \gamma$ be given. Then, with conditional probability (given $k$) at least $1 - \delta$, we have

$$\sigma_{(n-k)+n^\kappa+2} \leq \mathcal{O}\left(\sqrt{n^{\gamma+1}(\ln(1/\delta)+n)}\right) e^{-(n^{(\kappa-\gamma)})},$$

where the constant in $\mathcal{O}$ is universal.

*Proof.* Let $z_1, \ldots, z_k$ be the poles of $\overline{G}(z)$ inside $D(0, 1 - n^{-\gamma})$. Assume $G(z)$ can be written as

$$\overline{G}(z) = \overline{G}_1(z) + \sum_{i=1}^{k}\frac{c_i}{z - z_i},$$

where $\overline{G}_1(z)$ is a degree-$(n-k)$ rational function with poles inside the annulus of inner radius $1 - n^{-\gamma}$ and outer radius $1$ and $c_i$'s are i.i.d. random variables with distribution $\mathcal{N}(0, 1)$. We can further write

$$\sum_{i=1}^{k}\frac{c_i}{z - z_i} = \sum_{i=1}^{k}c_i\sum_{j=0}^{\infty}z_i^j z^{-j-1} = \sum_{j=1}^{\infty}z^{-1-j}\left(\sum_{i=1}^{k}c_i z_i^j\right).$$

By the AAK theory, for any $K > 0$, we have

$$\sigma_{(n-k)+K} \leq \sup_{|z|=1}\left|\sum_{j=K-2}^{\infty}z^{-1-j}\left(\sum_{i=1}^{k}c_i z_i^j\right)\right| \leq \sum_{j=K-2}^{\infty}\|\mathbf{c}\,\mathbf{z}^j\|_1 \leq \|\mathbf{c}\|_2\sum_{j=K-2}^{\infty}\|\mathbf{z}^j\|_2,$$

where $\mathbf{c} = [c_1, \ldots, c_k]^\top$ and $\mathbf{z}^j = [z_1^j, \ldots, z_k^j]^\top$, and the last step follows from Hölder's inequality. Since $\|\mathbf{c}\|_2^2$ follows the $\chi_k^2$ distribution, for $M > n$, we have that

$$P(\|\mathbf{c}\|_2^2 > M) \leq \exp\left(-\frac{n}{2}\left(\frac{M}{n} - 1 - \ln\left(\frac{M}{n}\right)\right)\right).$$

Hence, there exists a universal constant $C > 0$ such that when $M \geq C(\ln(1/\delta) + n)^3$, we have

$$P(\|\mathbf{c}\|_2^2 > M) \leq \exp\left(-\frac{n}{2}\frac{M}{n}\right) \leq \delta.$$

Moreover, since $\left|z_i^j\right| \leq (1 - n^{-\gamma})^j$ for all $1 \leq i \leq k$, we have

$$\|\mathbf{z}^j\|_2 \leq \sqrt{n}(1 - n^{-\gamma})^j.$$

That is, with probability at least $1 - \delta$, we have

$$\sigma_{(n-k)+K} \leq \mathcal{O}\left(\sqrt{\ln(1/\delta)+n}\right)\sqrt{n}\sum_{j=K-2}^{\infty}(1 - n^{-\gamma})^j = \mathcal{O}\left(\sqrt{n^{\gamma+1}(\ln(1/\delta)+n)}\right)(1-n^{-\gamma})^{K-2}.$$

Suppose $K \geq n^\kappa + 2$. Then, we have

$$(1-n^{-\gamma})^{K-2} \leq (1-n^{-\gamma})^{(n^\kappa)} = \left((1-n^{-\gamma})^{(n^\gamma)}\right)^{(n^{\kappa-\gamma})}.$$

Since $(1-n^{-\gamma})^{n^\gamma} \to e^{-1}$ as $n \to \infty$, if $\kappa > \gamma$, then $(1-n^{-\gamma})^{K-2}$ decays faster than any negative power of $n$. Hence, we have

$$\sigma_{(n-k)+K} \leq \mathcal{O}\left(\sqrt{n^{\gamma+1}(\ln(1/\delta)+n)}\right)e^{-(n^{(\kappa-\gamma)})},$$

as desired. $\qquad\square$

**Lemma 3.** With probability at least $1 - \delta$, the leading Hankel singular value $\sigma_1$ satisfies

$$\sigma_1 \geq \mathcal{O}(\sqrt{n}\delta),$$

where the constant in $\mathcal{O}$ is universal.

*Proof.* The leading Hankel singular value $\sigma_1$ is equivalent to the spectral norm of the Hankel matrix

$$\begin{bmatrix} \overline{\mathbf{CB}} & \overline{\mathbf{CAB}} & \overline{\mathbf{CA}^2\mathbf{B}} & \cdots \\ \overline{\mathbf{CAB}} & \overline{\mathbf{CA}^2\mathbf{B}} & \cdots & \cdots \\ \overline{\mathbf{CA}^2\mathbf{B}} & \vdots & \ddots & \vdots \\ \vdots & \vdots & \cdots & \ddots \end{bmatrix}.$$

Hence, we have

$$\sigma_1 \geq \left|\overline{\mathbf{CB}}\right|,$$

where $\overline{\mathbf{CB}}/\sqrt{n} \sim \mathcal{N}(0,1)$. Hence, if we set $M = C\sqrt{n}\delta$ for some universal constant $C > 0$, we have

$$P(\sigma_1 \leq M) \leq P\left(\left|\overline{\mathbf{CB}}\right| \leq M\right) = P\left(\left|\overline{\mathbf{CB}}\right|/\sqrt{n} \leq M/\sqrt{n}\right) \leq \delta.$$

This completes the proof. $\qquad\square$

Now, let's assemble our ultimate statement.

*Proof of Theorem 1.* By Lemma 2 and Lemma 3, with probability at least $1 - \delta/2$, we have that

$$\frac{\sigma_{(n-k+n^\kappa+2)}}{\sigma_1} \leq \mathcal{O}\left(\sqrt{n^\gamma(\ln(1/\delta)+n)}\delta^{-1}\right)e^{-(n^{(\kappa-\gamma)})}.$$

Hence, if we set $\kappa$ so that

$$n^\kappa \geq n^\gamma \ln\left(C\delta^{-1}\epsilon^{-1}\sqrt{n^\gamma(\log(1/\delta)+n)}\right) = n^\gamma \ln\left(\mathcal{O}\left(\delta^{-3/2}\epsilon^{-1}n^{(\gamma+1)/2}\right)\right)$$

---

[3]Here, $n$ is to guarantee that $M > n$

for a sufficiently large universal constant $C > 0$, then we guarantee that

$$\frac{\sigma_{(n-k+n^\kappa+2)}}{\sigma_1} \le \epsilon.$$

By Lemma 1, we have that with probability at least $1 - \delta/2$,

$$n - k \le n^\beta, \qquad \beta = 1 + \log_n\left(n^{-\gamma\alpha} + \sqrt{\frac{\ln(2/\delta)}{2n}}\right).$$

Set $\gamma = 1/(1 + \alpha)$. Then, we have

$$\beta = 1 + \log_n\left(n^{-\gamma\alpha}\left(1 + \sqrt{\frac{\ln(2/\delta)}{2n^{1-2\gamma\alpha}}}\right)\right) = 1 - \gamma\alpha + \log_n\left(1 + \sqrt{\frac{\ln(2/\delta)}{2n^{1-2\gamma\alpha}}}\right)$$

$$\le \frac{1}{1+\alpha} + \log_n(1 + \sqrt{\ln(2/\delta)/2}) < \frac{1}{1+\alpha} + \frac{\ln(2 + \sqrt{\ln(1/\delta)/2})}{\ln(n)}.$$

Since

$$n^\kappa + 2 = \mathcal{O}\left(n^\gamma \ln\left(\delta^{-3/2}\epsilon^{-1}n\right)\right) \le \mathcal{O}\left(n^\beta \ln\left(\delta^{-3/2}\epsilon^{-1}n\right)\right).$$

The claim is proved. □

## F    PROOF OF THEOREM 2

In this section, we prove Theorem 2. Our proof focuses on the worst-case perturbation by construction. As a consequence, it simultaneously proves the sharpness of the result. Intuitively, consider a rational function

$$s \mapsto \frac{bc}{s - a},$$

since we only care about its values on the imaginary axis, the closer the pole $a$ is to the imaginary axis, the less stable it is. On the other hand, it is obvious that the size of $bc$ also controls the (absolute) conditioning of the rational function. We state the rigorous proof below.

*Proof of Theorem 2.* Without loss of generality, we assume that $\mathbf{B} = \tilde{\mathbf{B}} = \begin{bmatrix} 1 & 1 & \cdots & 1 \end{bmatrix}^\top$.[4] The transfer functions of $\Gamma$ and $\tilde{\Gamma}$ are

$$G(s) = \sum_{j=1}^n \frac{c_j}{s - s_j} \qquad \text{and} \qquad \tilde{G}(s) = \sum_{j=1}^n \frac{\tilde{c}_j}{s - \tilde{s}_j},$$

respectively. Then, for any $s$ on the imaginary axis, we have

$$\left|\frac{c_j}{s - s_j} - \frac{\tilde{c}_j}{s - \tilde{s}_j}\right| = \left|\frac{c_j(s - \tilde{s}_j) - \tilde{c}_j(s - s_j)}{(s - s_j)(s - \tilde{s}_j)}\right| = \left|\frac{c_j s - c_j \tilde{s}_j - \tilde{c}_j s + \tilde{c}_j s_j + \tilde{c}_j \tilde{s}_j - \tilde{c}_j \tilde{s}_j}{(s - s_j)(s - \tilde{s}_j)}\right|$$

$$\le \frac{|c_j - \tilde{c}_j|\,|s - \tilde{s}_j| + |\tilde{c}_j|\,|s_j - \tilde{s}_j|}{|s - s_j|\,|s - \tilde{s}_j|} = \frac{|c_j - \tilde{c}_j|}{|s - s_j|} + \frac{|\tilde{c}_j|\,|s_j - \tilde{s}_j|}{|s - s_j|\,|s - \tilde{s}_j|}$$

$$\le \frac{\Delta_{\mathbf{B}}}{|\text{Re}(s_j)|} + \frac{2\,|c_j|\,\Delta_{\mathbf{A}}}{|\text{Re}(s_j)|^2/2}.$$

Hence, we have

$$\|G - \tilde{G}\|_\infty \le \sum_{j=1}^n \left(\frac{\Delta_{\mathbf{B}}}{|\text{Re}(s_j)|} + 4\frac{|c_j|\,\Delta_{\mathbf{A}}}{|\text{Re}(s_j)|^2}\right)$$

$$\le n\Delta_{\mathbf{B}} \max_j \frac{1}{|\text{Re}(s_j)|} + 4n\Delta_{\mathbf{A}} \max_j \frac{|c_j|}{|\text{Re}(s_j)|^2}.$$

---

[4]Otherwise, we can redefine $\mathbf{B} = \tilde{\mathbf{B}} = \begin{bmatrix} 1 & 1 & \cdots & 1 \end{bmatrix}^\top$, $\mathbf{C} = \mathbf{B}^\top \circ \mathbf{C}$, and $\tilde{\mathbf{C}} = \tilde{\mathbf{B}}^\top \circ \tilde{\mathbf{C}}$, and the redefined transfer functions are unchanged.

This proves the upper bound. To prove the lower bound, let $j_1$ be an index that maximizes $1/|\text{Re}(s_j)|$ and $j_2$ be an index that maximizes $|c_j|/|\text{Re}(s_j)|^2$. Define $\tilde{\Gamma}_{\mathbf{B}}$ by perturbing $c_{j_1}$ to $c_{j_1} + \Delta_{\mathbf{B}}$. Then, we have

$$\|G - \tilde{G}_{\mathbf{B}}\|_\infty = \left\|\frac{\Delta_{\mathbf{B}}}{s - s_{j_1}}\right\|_\infty = \Delta_{\mathbf{B}} \max_j \frac{1}{|\text{Re}(s_j)|}.$$

Define $\tilde{\Gamma}_{\mathbf{A}}$ by perturbing $s_{j_2}$ to $s_{j_2} + \Delta_{\mathbf{A}}$. Then, we have

$$\|G - \tilde{G}_{\mathbf{A}}\|_\infty = |c_{j_2}|\left\|\frac{1}{s - s_{j_2}} - \frac{1}{s - s_{j_2} + \Delta_{\mathbf{A}}}\right\|_\infty = |c_{j_2}|\Delta_{\mathbf{A}}\left\|\frac{1}{(s - s_{j_2})(s - s_{j_2} + \Delta_{\mathbf{A}})}\right\|_\infty$$

$$\geq |c_{j_2}|\Delta_{\mathbf{A}}\frac{1}{|\text{Re}(s_{j_2})|^2} = \Delta_{\mathbf{A}} \max_j \frac{|c_j|}{|\text{Re}(s_j)|^2}.$$

This proves the sharpness of the theorem. $\qquad\square$

Next, we show that the factor $n$ in Theorem 2 is in fact also tight.

**Proposition 1.** For any $n$, there exists a system $\Gamma$ of size $n$ and two systems $\tilde{\Gamma}_{\mathbf{A}}$ and $\tilde{\Gamma}_{\mathbf{B}}$ perturbed from $\Gamma$, with transfer functions $G$, $\tilde{G}_{\mathbf{A}}$, and $\tilde{G}_{\mathbf{B}}$, respectively, that satisfy the perturbation conditions in Theorem 2 and have

$$\|G - \tilde{G}_{\mathbf{A}}\|_\infty \geq n\Delta_{\mathbf{A}} \max_j \frac{|b_j c_j|}{|\text{Re}(a_j)|^2}, \qquad \|G - \tilde{G}_{\mathbf{B}}\|_\infty \geq n\Delta_{\mathbf{B}} \max_j \frac{1}{|\text{Re}(a_j)|}.$$

*Proof.* We take the single partial fraction from the proof of the lower bounds in Theorem 2 and repeat it for $n$ times to construct $\Gamma$. This would multiply the size of the transfer function perturbation by a factor of $n$. $\qquad\square$

## G   PROOF OF THEOREM 3 AND THEOREM 4

The proof of Theorem 3 is a straightforward assembly of two results in random matrix theory. The first result, due to Nekrutkin (2013), controls the Hankel norm $\sigma_1(\overline{\mathbf{H}}_n)$ of a random Hankel matrix, whereas the second result by Bryc et al. (2006) studies the distribution of all absolute singular values $\sigma_j(\overline{\mathbf{H}}_n)$ of a random Hankel matrix. Our study of the relative Hankel singular values is achieved by taking the quotient of the subjects of the two prior works.

*Proof of Theorem 3.* By Nekrutkin (2013), with probability 1, we have that

$$\sigma_1(\overline{\mathbf{H}}_n) = \|\overline{\mathbf{H}}_n\| = \mathcal{O}(\sqrt{n \ln n}).$$

Define $\overline{\mathbf{K}}_n = \overline{\mathbf{H}}_n(1:\lceil n/2\rceil, 1:\lceil n/2\rceil)$. Then, by Bryc et al. (2006), with probability 1, we have that $\mu(\overline{\mathbf{K}}_n/\sqrt{n})$ converges in distribution to a fixed probability measure, where

$$\mu(\overline{\mathbf{K}}_n/\sqrt{n}) = \frac{1}{\lceil n/2\rceil} \sum_{j=1}^{\lceil n/2\rceil} \delta_{\lambda_j(K_n/\sqrt{n})}$$

is the spectral measure of $\overline{\mathbf{K}}_n/\sqrt{n}$. Since $\overline{\mathbf{K}}_n$ is symmetric, the singular values of $\overline{\mathbf{K}}_n$ are the moduli of the eigenvalues of $\overline{\mathbf{K}}_n$. Hence, fix some $\epsilon > 0$, we have that

$$\left|\{j \mid \sigma_j(\overline{\mathbf{K}}_n)/\sigma_1(\overline{\mathbf{H}}_n) > \epsilon/\sqrt{\ln(n)}\}\right| = \Omega(n).$$

Since $\overline{\mathbf{K}}_n$ is a submatrix of $\overline{\mathbf{H}}_n$, we have $\sigma_j(\overline{\mathbf{H}}_n) \geq \sigma_j(\overline{\mathbf{K}}_n)$ for all $1 \leq j \leq \lceil n/2\rceil$. Hence, its $(\epsilon/\sqrt{\ln(n)})$-rank can be controlled as

$$\left|\{j \mid \sigma_j(\overline{\mathbf{H}}_n)/\sigma_1(\overline{\mathbf{H}}_n) > \epsilon/\sqrt{\ln(n)}\}\right| \geq \left|\{j \mid \sigma_j(\overline{\mathbf{K}}_n)/\sigma_1(\overline{\mathbf{H}}_n) > \epsilon/\sqrt{\ln(n)}\}\right| = \Omega(n).$$

This finishes the proof. $\qquad\square$

We remark that in the statement of Theorem 3, we study the $\epsilon/\sqrt{\ln(n)}$-rank of the Hankel matrix instead of the $\epsilon$-rank. The reason is that, unlike the singular values of a random matrix, the spectral measure of a normalized random Hankel matrix $\mathbf{H}_n/\sqrt{n}$ has unbounded support. In order to pull the $1/\sqrt{\ln(n)}$ factor out and make it into the $\Omega(n)$ bound, we need to study the distribution of the spectral measure of $\mathbf{H}_n/\sqrt{n}$. As pointed out by Bose (2018), however, this seems to be a hard problem. Nevertheless, we can empirically test the statement by numerical experiments.

Next, we provide the proof of Theorem 4, which is almost immediate from Hölder's inequality.

*Proof of Theorem 4.* By Hölder's inequality, we have

$$\|G - \tilde{G}\|_\infty = \|\overline{G} - \overline{\tilde{G}}\|_\infty \le \max_{|z|=1} \sum_{j=0}^{n-1} |\mathbf{h}_j - \tilde{\mathbf{h}}_j||z|^{-j-1} \le \|\mathbf{h} - \tilde{\mathbf{h}}\|_2 \sqrt{n}. \qquad \square$$

# H SOME NUMERICAL EXPERIMENTS ON HANKEL MATRICES

## H.1 NUMERICAL RANKS OF RANDOM LTI SYSTEMS AND RANDOM HANKEL MATRICES IN PRACTICE

While the theoretical part of our paper focuses on the $\epsilon$-rank of an LTI system, in the main text, we showed the distribution of all Hankel singular values of different LTI systems. The main reason for showing the histograms instead of a single number (i.e., the $\epsilon$-rank) is that the histogram gives us more information while the $\epsilon$-rank is merely a cutoff. In this section, we empirically compute the $\epsilon$-rank to verify the two theorems (i.e., Theorem 1 and Theorem 3).

In this experiment, we always set $\epsilon = 0.01$. For every $n$ in our experiment, we first randomly initialize a random $n \times n$ Hankel matrix

$$\overline{\mathbf{H}}_n = \begin{bmatrix} h_0 & h_1 & h_2 & \cdots & h_{n-1} \\ h_1 & h_2 & \cdots & h_{n-1} & 0 \\ h_2 & \cdots & h_{n-1} & 0 & 0 \\ \vdots & \cdot{}^{\cdot{}^{\cdot}} & \cdot{}^{\cdot{}^{\cdot}} & \vdots & \vdots \\ h_{n-1} & 0 & \cdots & 0 & 0 \end{bmatrix},$$

where $h_j$ are i.i.d. random Gaussian variables with variance of 1. We compute its $\epsilon$-rank and we repeat the experiment for 1000 trials. Similarly, for every $n$, we randomly initialize an LTI system with

$$\mathbf{A} = \mathrm{diag}(a_1, \ldots, a_n), \qquad a_j \sim \mathrm{Uniform}(\mathbb{D}),$$

where $\mathbb{D}$ is the open unit disk in the complex plane and the elements of $\mathbf{B} \circ \mathbf{C}^\top$ are sampled i.i.d. from $\mathcal{N}(0, 1)$. We compute its $\epsilon$-rank and also repeat the experiment for 1000 trials. From Figure 8, we see that a random LTI system has a low rank, whereas a random Hankel matrix has a high rank in the sense that it is about proportional to $n$. This observation aligns with our theory in Theorem 1 and Theorem 3,

## H.2 HANKEL MATRICES ARE STABLE TO PERTURBATION IN PRACTICE

Theorem 4 predicts that the Hankel matrices are very stable when being perturbed. In this section, we run experiments in parallel with those in Figure 4, where we perturb a Hankel matrix $\overline{\mathbf{H}}$. As in Figure 4, we also set the size of the random perturbation to be 1% and 0.1%, respectively, of the original system. Theorem 4 is corroborated by Figure 9, where we see that a small perturbation has a minimal effect on the Hankel singular values.

## H.3 HANKEL SINGULAR VALUES AFTER TRAINING

In the main text, we showed the distributions of the Hankel singular values at initialization and after 10 epochs. In this section, we show the distribution of the Hankel singular values after the training is done.

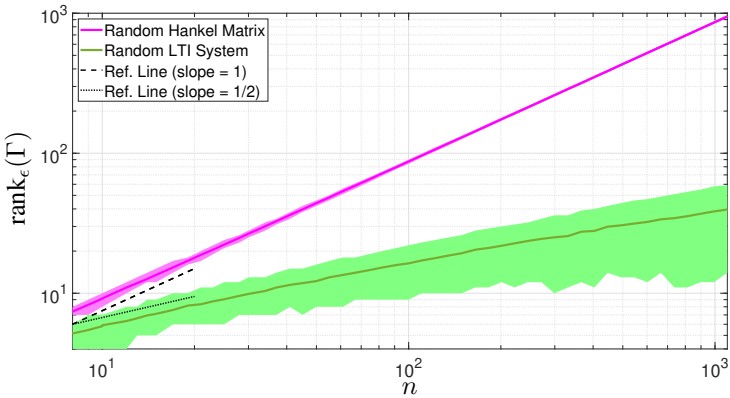

**Figure 8:** The $\epsilon$-rank of a random LTI system of size $n$, where $\epsilon = 0.01$. The random systems are parameterized by a Hankel matrix or by the matrices $\mathbf{A}, \mathbf{B}$, and $\mathbf{C}$. The lines are the average rank and the shaded regions indicate the $10\%$-$90\%$ range over 1000 trials.

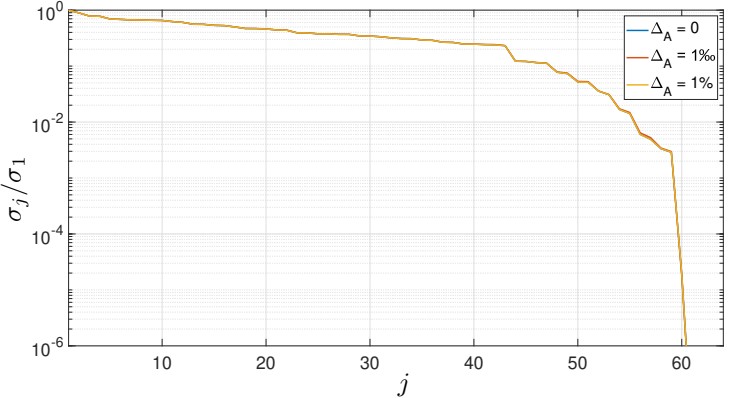

**Figure 9:** A random perturbation is added to $\overline{\overline{\mathbf{H}}}$. The magnitude of the perturbation is set to $0.1\%$ and $1\%$ of the original matrix. We show the relative Hankel singular values $\sigma_j/\sigma_1$ of the original and perturbed systems. In this case, the three curves are almost overlapping.

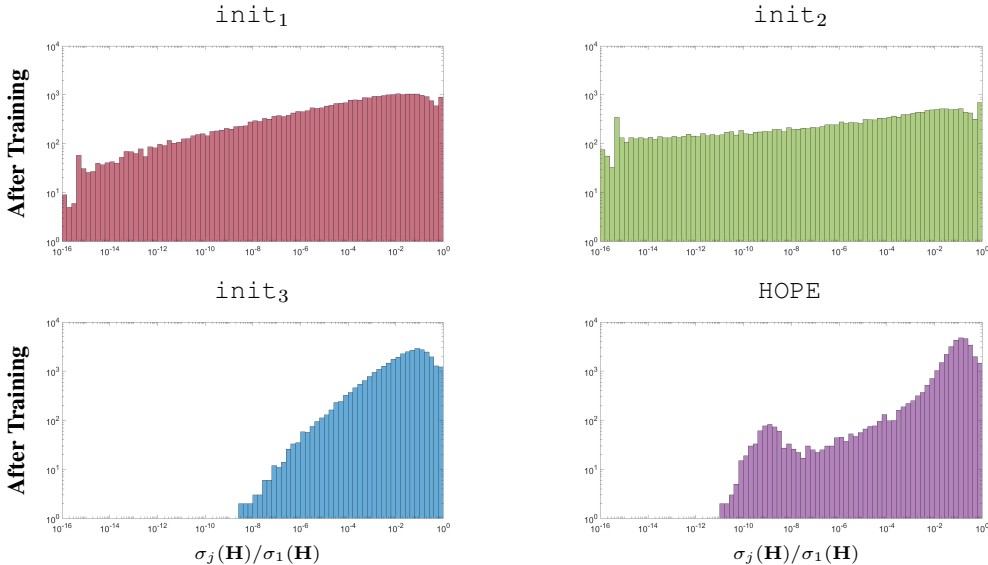

**Figure 10:** The distribution of Hankel singular values after training.

# I    ADJUSTING THE DISCRETIZATION STEP VIA NUFFT

In this section, we derive Algorithm 1. If we set $\Delta t = 1$ and discretize $\Gamma$, the convolutional kernel is exactly $\mathbf{h}$ padded with $L - n$ zeros, where $L$ is the length of the sequential input. However, we usually want to discretize with a different $\Delta t$. We can do so via resampling the transfer functions of $\Gamma$ and $\overline{\Gamma}$, which equal (Peller, 2003)

$$G(s) = \sum_{j=0}^{n-1} \mathbf{h}_j \left( (1+s)/(1-s) \right)^{-j-1} \qquad \Leftrightarrow \qquad \overline{G}(z) = \sum_{j=0}^{n-1} \mathbf{h}_j z^{-j-1}. \qquad (11)$$

Let $\boldsymbol{\omega}^{(L)} = \begin{bmatrix} \omega_0^{(L)} & \cdots & \omega_{L-1}^{(L)} \end{bmatrix}^\top$ be the vector of the $L$th roots of unity. Given an input $\mathbf{u} \in \mathbb{C}^{L \times 1}$ of length $L$, when $\Delta t = 1$, the outputs can be evaluated as

$$\mathbf{y} = \texttt{iFFT}(\texttt{FFT}(\mathbf{u}) \circ \overline{G}(\boldsymbol{\omega}^{(L)})), \qquad \overline{G}(\boldsymbol{\omega}^{(L)}) = \begin{bmatrix} \overline{G}(\omega_0^{(L)}) & \cdots & \overline{G}(\omega_{L-1}^{(L)}) \end{bmatrix}^\top.$$

For a different $\Delta t$, one way to think of it is that we have compressed or dilated the time domain of $\mathbf{u}$ by a factor of $\Delta t$. Hence, the frequency domain of its Fourier transform $\hat{\mathbf{u}}$ is dilated or compressed by a factor of $1/\Delta t$. That is, we should relocate our samplers in the frequency domain as

$$\boldsymbol{\omega}^{(L, \Delta t)} = \begin{bmatrix} \omega_0^{(L, \Delta t)} & \cdots & \omega_{L-1}^{(L, \Delta t)} \end{bmatrix}^\top, \qquad \omega_j^{(L, \Delta t)} = \frac{1 + s_j^{(L)}/\Delta_t}{1 - s_j^{(L)}/\Delta_t}, \qquad s_j^{(L)} = \frac{\omega_j^{(L)} - 1}{\omega_j^{(L)} + 1},$$
$$(12)$$

where $s_j^{(L)}/\Delta t$ and $\omega_j^{(L, \Delta t)}$ are the scaled samplers in the time and angular domain, respectively. Then, the output sequence $\mathbf{y}$ can be computed, from the nonuniform samples $\overline{G}(\boldsymbol{\omega}^{(L, \Delta t)})$, as

$$\mathbf{y} = \texttt{iFFT}(\texttt{FFT}(\mathbf{u}) \circ \overline{G}(\boldsymbol{\omega}^{(L, \Delta t)})), \qquad \overline{G}(\boldsymbol{\omega}^{(L, \Delta t)}) = \begin{bmatrix} \overline{G}(\omega_0^{(L, \Delta t)}) & \cdots & \overline{G}(\omega_{L-1}^{(L, \Delta t)}) \end{bmatrix}^\top. \quad (13)$$

To understand why eq. (12) holds, assume there exists a continuous function $u$ on the unit circle $\partial \mathbb{D}$ where the discrete inputs $\mathbf{u}$ are sampled from. Then, FFT allows us to write $u$ into the Fourier expansion:

$$u(z) = \sum_{j=0}^{L-1} [\texttt{FFT}(\mathbf{u})]_j \exp\left( -2\pi i \frac{j}{L} z \right).$$

By the property of the transfer function eq. (3), we know that the output function $y$ is equal to

$$y(z) = \sum_{j=0}^{L-1} \underbrace{\left( [\texttt{FFT}(\mathbf{u})]_j \, \overline{G}(\omega_j^{(L)}) \right)}_{\hat{\mathbf{y}}_j} \exp\left( -2\pi i \frac{j}{L} z \right).$$

To compute the discrete output $\mathbf{y}$, one samples $y$ at $z = \omega_0^{(L)}, \ldots, \omega_{L-1}^{(L)}$, which is equivalent to an inverse FFT on $\texttt{FFT}(\mathbf{u}) \circ \overline{G}(\boldsymbol{\omega}^{(L)})$.

Now, if we want to change $\Delta t$, one way to think of it is as if our LTI system is unchanged, but the time domain of $u(z)$ is scaled by a factor of $\Delta t$. That is, we now have[5]

$$u^{(\Delta t)}(z) = \sum_{j=0}^{L-1} [\texttt{FFT}(\mathbf{u})]_j \exp\left( -2\pi i \frac{j}{L} z^{(\Delta t)} \right), \qquad z^{(\Delta t)} = \frac{1 + s/\Delta_t}{1 - s/\Delta_t}, \qquad s = \frac{z-1}{z+1}.$$

The output function $y^{(\Delta t)}$ is now equal to

$$y^{(\Delta t)}(z) = \sum_{j=0}^{L-1} \left( [\texttt{FFT}(\mathbf{u})]_j \, \overline{G}(\omega_j^{(L, \Delta t)}) \right) \exp\left( -2\pi i \frac{j}{L} z^{(\Delta t)} \right).$$

---

[5]Note that we could alternatively scale the angular domain instead of the time domain, i.e., $z^{(\Delta t)} = z/\Delta t$. The difference is on the level of discretization. However, we find that discretizing the time domain gives us a better performance in general.

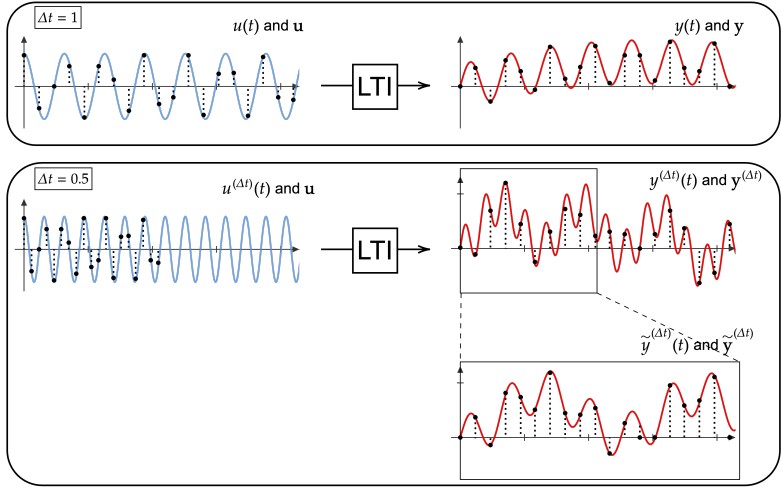

**Figure 11:** A visualization of appendix I.

The only real difficulty is that when we sample $y^{(\Delta t)}$ at $z = \omega_0^{(L)}, \ldots, \omega_{L-1}^{(L)}$ to obtain $\mathbf{y}^{(\Delta t)}$, we note that $z^{(\Delta t)} = \omega_0^{(L,\Delta t)}, \ldots, \omega_{L-1}^{(L,\Delta t)}$ are not uniform on the unit circle. Hence, it cannot be achieved via inverse FFT. However, this sampling can be done via the so-called nonuniform FFT (NUFFT), which also takes $\mathcal{O}(L \log L)$. In general, one can interpret FFT as the procedure of evaluating a function

$$f(\omega) = \sum_{j=0}^{n-1} f_j \exp\left(-j\omega\right),$$

at the degree-$L$ roots of unity $\boldsymbol{\omega} = (\omega_0, \ldots, \omega_{L-1})$. The NUFFT is the procedure of evaluating exactly the same function $f$, but at potentially nonuniform nodes $\tilde{\boldsymbol{\omega}} \neq \boldsymbol{\omega}$. We remark that NUFFT is only used to simplify the representation of our derivation of Algorithm 1. It is not explicitly used in the algorithm, even though the numerical stability of the NUFFT procedure is studied in Barnett (2022); Yu & Townsend (2023); Austin (2023), and fast algorithms can be found in Greengard & Lee (2004); Barnett (2022); Wilber et al. (2024).

Using the NUFFT at the uneven samples $\boldsymbol{\omega}^{(L,\Delta t)}$, we obtain

$$\mathbf{y}^{(\Delta t)} = \text{NUFFT}(\text{FFT}(\mathbf{u}) \circ \overline{G}(\boldsymbol{\omega}^{(L,\Delta t)})).$$

Now, consider the following function

$$\tilde{y}^{(\Delta t)}(z) = y^{(\Delta t)}(z^{(1/\Delta t)}) = \sum_{j=0}^{L-1} \left([\text{FFT}(\mathbf{u})]_j \, \overline{G}(\omega_j^{(L,\Delta t)})\right) \exp\left(-2\pi i \frac{j}{L} z\right).$$

This output function is a scaled version of $y^{(\Delta t)}$, where we scale the time domain by a factor of $1/\Delta t$. One can sample $\tilde{y}^{(\Delta t)}$ at $z = \omega_0^{(L)}, \ldots, \omega_{L-1}^{(L)}$ using iFFT:

$$\tilde{\mathbf{y}}^{(\Delta t)} = \text{iFFT}(\text{FFT}(\mathbf{u}) \circ \overline{G}(\boldsymbol{\omega}^{(L,\Delta t)})).$$

This leads us to eq. (13).

## J  CONVERGENCE OF HOPE

Assume that there exists a target dynamical system $\overline{\Gamma}_* = (\overline{\mathbf{A}}_*, \overline{\mathbf{B}}_*, \overline{\mathbf{C}}_*)$ that we would like to learn using a HOPE system parameterized by $\mathbf{h}$, where our objective is to minimize the $L^2$-loss of the transfer functions. Let $\overline{G}_*$ and $\overline{G}_{\mathbf{h}}$ be the transfer functions of the target system and the HOPE

system, respectively. This least-square system can be written as follows:

$$\underbrace{\begin{bmatrix} | & | & & | \\ z^{-1} & z^{-2} & \cdots & z^{-n} \\ | & | & & | \end{bmatrix}}_{\mathbf{Z}} \begin{bmatrix} \mathbf{h}_1 \\ \mathbf{h}_2 \\ \vdots \\ \mathbf{h}_n \end{bmatrix} = \begin{bmatrix} | \\ \overline{G}_* \\ | \end{bmatrix},$$

where $\mathbf{Z}$ is a quasimatrix. Hence, optimizing $\mathbf{h}$ is a convex problem. Moreover, the columns of $\mathbf{Z}$ are orthogonal!

**Proposition 2.** Suppose we apply gradient descent on $\mathbf{h}$ with learning rate $\eta < 1$ to minimize $\|\overline{G}_{\mathbf{h}} - \overline{G}_*\|_2$, where the 2-norm is taken over the unit circle. Let $\mathbf{h}^{(k)}$ be the parameter after the $k$-th iteration. Then, we have

$$\|\overline{G}_{\mathbf{h}^{(k+1)}} - \overline{G}_*\|_2 \leq (1-\eta)\|\overline{G}_{\mathbf{h}^{(k)}} - \overline{G}_*\|_2 \leq \cdots \leq (1-\eta)^{k+1}\|\overline{G}_{\mathbf{h}^{(0)}} - \overline{G}_*\|_2.$$

*Proof.* The proof is immediate from convex optimization theory by noting that the condition number $\kappa_2(\mathbf{Z}) = 1$. $\qquad\square$

In our setting, we optimize on a continuous level. We can also analyze the optimization on a discrete level, where one takes the samples of $\overline{G}_*$ and $\overline{G}_{\mathbf{h}}$. In that case, the matrix $\mathbf{Z}$ is no longer a quasimatrix but instead a true NUFFT matrix. Its conditioning has been widely studied in the literature (see (Yu & Townsend, 2023; Barnett, 2022)), making analogues of Proposition 2 easy to derive.

## K    EXPERIMENTAL DETAILS

In this section, we provide the details of the three experiments presented in the main text.

### K.1    ANALYZING HANKEL SINGULAR VALUES USING THE SCIFAR-10 TASK

As mentioned in section 3, in the experiments presented in Figure 2 and Figure 6, we always train an SSM with 4 layers and 128 channels. Each channel in a layer is modeled by an LTI system with $n = 64$ states. When we parameterize the LTI systems using $\mathbf{A}, \mathbf{B}, \mathbf{C}$, and $\mathbf{D}$, we assign a learning rate of $0.001$ to $\mathbf{A}$ and of $0.01$ to the rest. When we freeze the system matrices, then we set the learning rate of $\mathbf{A}, \mathbf{B}$ and $\mathbf{C}$ to $0$ while keeping that of $\mathbf{D}$ to be $0.01$. Note that the matrix $\mathbf{D}$ does not affect the Hankel singular values. All other model parameters are trained with a learning rate of $0.01$. For an LTI system parameterized by the Hankel matrix $\overline{\mathbf{H}}$, we adopt the same setting, except that $\overline{\mathbf{H}}$ is trained with a non-reduced learning rate of $0.01$. To compute the Hankel singular values of an LTI system $(\mathbf{A}, \mathbf{B}, \mathbf{C}, \mathbf{D})$, we use its balanced realization (see Appendix B). To compute the Hankel singular values of a system parameterized by $\overline{\mathbf{H}}$, we apply an SVD to the matrix $\overline{\mathbf{H}}$.

### K.2    TESTING HOPE-SSMS LONG MEMORY USING NOISY-SCIFAR

In this experiment (see Figure 7), we modify the sequential CIFAR-10 dataset by padding random sequences to the right. For each sequence of length $1024$ from the original dataset, we pad another sequence of length $1024$ to the end of it. The entries are sampled independently from the Gaussian distribution on the same magnitude as the entries in the original sequences. We adopt the same model architectures and learning rates as described in Appendix K.1 but make two exceptions. First, we fix the discretization size to be $\Delta t = 0.1$, and therefore, it does not need a learning rate. In addition, a canonical SSM decodes the output sequence by first doing a pooling. Here, instead of pooling over all $2048$ output vectors, to make the problem more challenging and require longer memory, we only pool over the last $1024$ output vectors. These correspond to the output vectors when the noises are fed. We also test the models using different discretization sizes $\Delta t$. We see that when $\Delta t = 1$, the S4D model fails to converge while our HOPE-SSM performs relatively well; on the other hand, when $\Delta t = 0.01$, both models tend to have a relatively good performance. These observations align with our theory because a larger $\Delta t$ means that we put the discrete data on a continuous time domain with a longer span; hence, longer memory capacity is needed.

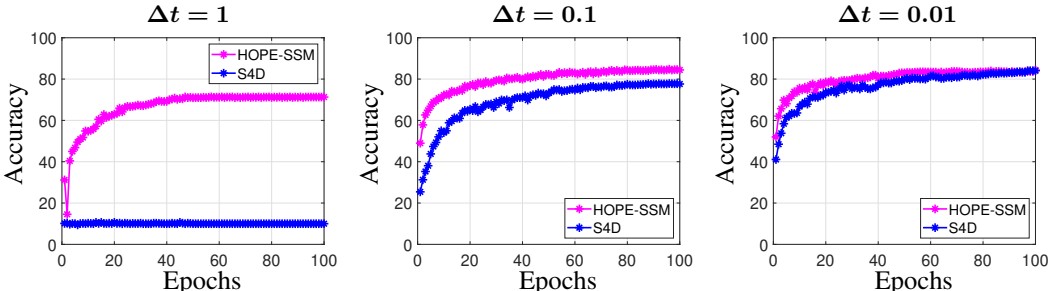

**Figure 12:** Performance of the HOPE-SSM and the S4D model on the noise-padded sCIFAR-10 task using different values of $\Delta t$.

### K.3 HYPERPARAMETERS OF HOPE-SSMS IN THE LONG-RANGE ARENA

In this section, we present the table of hyperparameters used to train our HOPE-SSM on the LRA tasks (Tay et al., 2021) (Apache License, Version 2.0). (See Table 2.) Our codes are adapted from the code associated with the original S4 and S4D papers (Gu et al., 2022b;a) (Apache License, Version 2.0). Note that compared to the hyperparameters used to train S4 and S4D, we use the same model hyperparameters and only slightly tune the training hyperparameters. All experiments are done on a NVIDIA A30 Tensor Core GPU with 24 GB of memory. The time efficiency of our model is roughly the same as that of the S4D model.

| Task | Depth | #Features | Norm | Prenorm | DO | LR | BS | Epochs | WD | $\Delta$ Range |
|---|---|---|---|---|---|---|---|---|---|---|
| ListOps | 8 | 256 | BN | False | 0. | 0.01 | 20 | 100 | 0.03 | (0.001,0.1) |
| Text | 6 | 256 | BN | True | 0.01 | 0.01 | 16 | 150 | 0.05 | (0.001,0.1) |
| Retrieval | 6 | 128 | BN | True | 0. | 0.008 | 32 | 80 | 0.03 | (0.001,0.1) |
| Image | 6 | 128 | LN | False | 0.1 | 0.004 | 32 | 1500 | 0.01 | (0.001,10) |
| Pathfinder | 6 | 256 | BN | True | 0. | 0.001 | 16 | 250 | 0.03 | (0.0001,0.1) |
| Path-X | 6 | 128 | BN | True | 0. | 0.001 | 16 | 100 | 0.04 | (0.0001,1) |

**Table 2:** Configurations of the HOPE-SSM model, where DO, LR, BS, and WD stand for dropout rate, learning rate, batch size, and weight decay, respectively.

### K.4 FURTHER ABLATION

We study the role of $\Delta t$ in the discretization of a HOPE-SSM. Here, we show that it is key to have tunable $\Delta t$ in the model of HOPE. This justifies our continuous-level parameterization of the convolutional kernel.

**Table 3:** An ablation of the HOPE-SSM that shows $\Delta t$ is essential for a good performance of the model.

| Model | ListOps | Text | Retrieval | Image | Pathfinder | Path-X | Avg. |
|---|---|---|---|---|---|---|---|
| HOPE-SSM | 62.60 | 89.83 | 91.80 | 88.68 | 95.73 | 98.45 | 87.85 |
| HOPE-SSM ($\Delta t = 1$) | 55.50 | 86.63 | 86.12 | 79.92 | 91.20 | 50.00 | 74.90 |

## L SUPPLEMENTARY EXPERIMENTS: RECENCY BIAS OF HOPE

The HOPE parameterization does not introduce a natural inductive bias over the inputs that fit into the $[0, n]$ memory window (see Figure 5). In some cases, however, a recency bias is desired to solve the tasks. In this section, we discuss how the recency bias can be tuned in our model. The first method is to change $\Delta t$ to ask for more or less bias. That is, if $\Delta t$ is set to be larger, then fewer inputs can be fit into the memory window $[0, n]$, so recency bias is advocated. We also propose a different way to tune the bias by introducing an explicit decay to the Markov parameters. That is, instead of setting $\overline{\mathbf{H}}_{ij} = \mathbf{h}_{i+j}$, we use $\overline{\mathbf{H}}_{ij} = c_{i+j}\mathbf{h}_{i+j}$ for some $c_j \to 0$ as $j \to \infty$. For example,

$c_j = (1+j)^\alpha$ for some hyperparameter $\alpha < 0$. This would introduce a natural "memory decay" over the interval $[0, n]$ and hence tune the bias.

We now use a synthetic experiment to show that we can tune the recency bias. Our datasets contain input sequences of length 1000, where the first 990 entries are sampled from i.i.d. random Gaussian distributions, and the last 10 are sampled from $c\cos(st)$ for some random $c \in [0, 1]$ and $s \in [0, 1]$. The goal is to predict the frequency $s$ given the noisy input. We train our model using different $\Delta t$ and the $\alpha$ hyperparameters. This task clearly requires recency bias because only the recent 10 inputs are relevant to the output.

**Table 4:** The mean-squared error in predicting the frequency $s$ given different training hyperparameters.

| | | $\Delta t$ | | | | | |
|---|---|---|---|---|---|---|---|
| | | **0.1** | **1** | **5** | **10** | **50** | **100** |
| | **0** | 0.084 | 0.053 | 0.020 | 0.006 | 0.006 | 0.015 |
| | **-0.25** | 0.053 | 0.046 | 0.022 | 0.004 | 0.008 | 0.015 |
| | **-0.5** | 0.055 | 0.048 | 0.021 | 0.008 | 0.013 | 0.016 |
| $\alpha$ | **-0.75** | 0.050 | 0.027 | 0.025 | 0.010 | 0.014 | 0.016 |
| | **-1** | 0.047 | 0.025 | 0.023 | 0.009 | 0.014 | 0.017 |
| | **-1.5** | 0.041 | 0.018 | 0.022 | 0.013 | 0.017 | 0.020 |
| | **-2** | 0.042 | 0.022 | 0.024 | 0.021 | 0.019 | 0.023 |

We observe that as $\Delta t$ increases and $\alpha$ gets more negative, the model indeed favors the recent inputs. In this experiment, we set $n = 64$. Hence, about $64/\Delta t$ inputs can be fit into the memory window. setting $\Delta t = 10$ allows the noises to be ignored by the model. In Table 4, we indeed see that it leads to the best performance of the model. We remark that since our setting requires a hard-threshold recency bias (i.e., we ideally want equal memory on only the last 10 inputs), it is more effective to tune the recency bias by changing $\Delta t$ than $\alpha$. However, the first two columns indeed show that we can tune the bias also by changing $\alpha$. Changing $\alpha$ could be more effective on other tasks that require a soft recency bias.

