# OpenReview forum: "HOPE for a Robust Parameterization of Long-memory State Space Models"
_ICLR.cc/2025/Conference — ICLR 2025 Poster_

### Official Review · Reviewer_TW3G · 2024-10-30

**Soundness:** 2
**Presentation:** 4
**Contribution:** 2
**Rating:** 6
**Confidence:** 4

**Summary:**

This paper proposes an alternative parameterisation of the state space models, where the SSM layer's dynamics is replaced by parameterising rows of the Hankel matrix. It is shown that classical SSM parameterisation tends to give rise to Hankel matrices with fast decaying Hankel singular values, and is posited to correlate with poor performance for long memory tasks. The proposed parameterisation does not involve repeated multiplication of the state space matrix, hence avoiding this shortcoming. These are supported by theoretical results, and the proposed method is shown to improve upon existing SSM variants on sCIFAR and long range arena.

**Strengths:**

While the paper's identified limitation (i.e. favoring exponential decay in memory)  of SSM is widely known and studied, its use of the Hankel numerical rank to study it is interesting. The theoretical results are carefully derived and the practical results show good promise. Finally, the paper is carefully written and the explanations of their approach and results are excellent.

**Weaknesses:**

The main weakness, to my understanding, is the premise of the proposed approach. In classical LTI and SSM literature on system identification (e.g. Tangirala, A. K. *Principles of System Identification: Theory and Practice*), the very reason why SSMs were introduced in favor of convolutional models is its ability to handle infinite sequences with a finite parameterisation. The present approach appears to go in the reverse direction:

1. In particular, I cannot see what is the difference between HOPE-SSM and the classical convolutional model (the most basic sequence to sequence model in LTI literature, where one directly parameterises the impulse response function), with the filter size $n$ is chosen as the same the original SSM's hidden state dimension. In fact,  the parameterisation (5) via $h$ seems to be exactly the convolution filter (impulse response), since each row of the Hankel matrix is just the impulse response (shifted by 1 for each subsequent row). If this were the case, the method has little to do with Hankel matrix, but is rather just a convolutional model based on parameterising the impulse response function.
2. The HOPE-SSM model's size scales with sequence length if it must process long sequences. In particular, all memory functions that can be represented are finitely supported, so it is not accurate to say that this improves upon SSM parameterisation, which is used in preference to convolutional models *precisely* because it allows the parameterisation of memory of infinite support with finite number of parameters (albeit introducing curse of memory due to forcing an exponential decay, as observed in previous literature).
3. The observed strengths of HOPE-SSM (non-exponential decay) are also present in convolutional networks (e.g. WaveNet with small filters but multiple layers, or single layer direct parameterisation of the impulse response), thus the numerical comparison should be made with those models, instead of just SSM variants.

**Questions:**

1. Can the authors clarify what is the fundamental difference, if any, of the HOPE-SSM and classical convolutional models that directly parameterises a truncated impulse response function? Looking at Alg 1, the only possible departure appears to be in the for loop in steps 6-7.
2. Can the authors clarify how (or if) the proposed model can process long sequences without increasing the filter size, while still meaningfully extract memory structures?
3. If different, can the authors comment and demonstrate both theoretically and numerically their advantage against convolutional-type sequence models (e.g. WaveNet/temporal CNNs, and directly paramterising the impulse response function)?
4. Is the conclusion of Theorem 1 verified by results in Fig 3 quantitatively?

Post rebuttal/revision: Appreciate the comparison with discrete convolutional models to assess the model's performance. I think while the novelty is still limited (it is at most a rescaled temporal convolution model with large filters), the performance appears good and the paper is well written. I have increased my score.

---

> ### Author Response · Authors · 2024-11-19
> **Response to the Reviewer**
>
> We thank the reviewer for the careful review and insightful comments. Below we address the questions and comments raised in this review.
>
> * W1: Thank you for raising this good point. We are aware that in our paper, we did not directly explain the difference between our method and traditional convolutional methods. We have expanded section 4 (see the texts in red and Figure 5) to explain this difference in detail. In particular, HOPE does not parameterize the discrete convolutional kernel (i.e., the discrete impulse response). Instead, it parameterizes a continuous kernel, which will be discretized **with respect to a different $\Delta t$**. That means our method can handle inputs with varying lengths. Moreover, its memory window is not just the past $n$ inputs; instead, its memory window is the past $n / \Delta t$ inputs with a trainable $\Delta t$. We hope this elucidates the key difference between our method and a convolutional model.
>
> * W2: We believe that this weakness can also be answered by our explanation above. Due to the $\Delta t$ parameter, the size of the memory support is not the constant $n$, but $n / \Delta t$ instead. In addition, in section 4 (Advantage III), we have acknowledged that the memory gain of HOPE is not asymptotic but rather empirical.
>
> * W3: We have done some further ablation studies to compare HOPE to a convolution-type model. That is, we directly parameterize the discrete convolutional kernel. Results are shown below on Long-Range Arena tasks (also added as Table 3 in the manuscript), which demonstrate the benefit of our continuous-level parameterization. Here, "x" means the model cannot beat random guessing.
>
>    |         | ListOps | Text | Retrieval  | Image | PathFinder | PathX |
>    |:-------:|:-------:|:-------:|:-------:|:-------:|:-------:|:-------:|
>    |Convolutional Model|55.50|86.63|86.12|79.92|91.20|x|
>    |HOPE-SSM|62.60|89.83|91.80|88.68|95.73|98.45|
>
> * Q1: Please find our response to Weakness 1 for an explanation of the difference between our model and a convolutional one.
>
> * Q2: Fixing an $n$, as the sequence length increases, one can decrease $\Delta t$ to fit a larger amount of input history into the memory window. This is exactly what we have done in the PathX case, where the sequence length is $16384$ and $n = 64$ in our model (see Table 2).
>
> * Q3: Please see our response to Weakness 3 for an empirical comparison between HOPE and a traditional convolutional model. Given our discussion of the continuous-time parameterization, the theoretical benefit of HOPE over a convolutional model is then clear: a convolutional model has a fixed memory window only depending on $n$ (as the reviewer has correctly identified) whereas the memory window of HOPE depends also on $\Delta t$ and can change during training.
>
> * Q4: Theorem 1 is an asymptotic analysis as $n \rightarrow \infty$; since Figure 3 has a fixed $n$, it cannot directly verify the theorem. However, the green histograms correspond to the assumptions made in Theorem 1 with $\alpha = 1$, and one can see that a lot of Hankel singular values are small. (Note that the histograms are on the log-log scale, so the green histograms indicate a rapid decay of the singular values.) We nevertheless numerically verify Theorem 1 (and Theorem 3) in Figure 8, where we sample random LTI systems with varying $n$ and compute their $\epsilon$-ranks.
>
> We hope this answers the reviewer's questions and concerns. We are happy to answer any follow-up question(s) that the reviewer may have.

---

> > ### Comment · Reviewer_TW3G · 2024-11-23
> > **Thanks for the replies and some further clarifications**
> >
> > Thank you for the response and the update. The comparison to the discrete convolution without time-rescaling parameter is useful.
> >
> > I'd like to further clarify regarding Q1/W1: under HOPE-SSM which now I understand is a convolutional model but whose temporal discretisation step can be varied (I still do not see how this is related to the Hankel matrix, it is still a convolutional model).
> >
> > Let's take a finite input sequence $x$ (say of length n), and consider now two input sequences
> > 1. x padded with $L-n$ zeros, forming a sequence of length $L$
> > 2. x padded with $2L-n$ zeroes, forming a sequence of length $2L$
> > Now, under the usual convolution model, as long as the filter size $h$ is greater than $n$, they will process both inputs identically, giving the same outputs. However, if I understand correctly, HOPE-SSM will give different outputs because the temporal domains are scaled according to the input sequence length. Is this correct? If so, would one foresee some issues with scaling time in this manner?

---

> ### Author Response · Authors · 2024-11-23
> **Thank you for your reply and response to follow-up questions**
>
> Dear reviewer,
>
> Thank you for reading through our response and raising the follow-up question(s). We identify two sub-questions in your comment, which we address below:
>
> 1. **Difference between HOPE and convolutional models:** Our HOPE model can indeed be computed by a convolution in the end. On the other hand, this is true for all state-space models. (See the original SSM paper [1], for example, where the model is computed by convolution.) Another way to view the main difference between different models, in that sense, is the way that the discrete convolutional kernel is parameterized. In a traditional convolutional NN, the discrete kernel (of a fixed length) is directly parameterized; in a canonical SSM, the convolutional kernel is parameterized by $(\mathbf{A}, \mathbf{B}, \mathbf{C})$ and $\Delta t$, and the $k$th entry of the kernel is then computed as $\overline{\mathbf{C}} \overline{\mathbf{A}}^k \overline{\mathbf{B}}$, where $(\overline{\mathbf{A}}, \overline{\mathbf{B}}, \overline{\mathbf{C}})$ is obtained by discretizing $(\mathbf{A}, \mathbf{B}, \mathbf{C})$ with respect to $\Delta t$; the HOPE model, on the other hand, parameterizes the kernel using $\mathbf{h}$ and $\Delta t$, and the kernel is obtained by continuating $\mathbf{h}$ to the entire time domain $[0,\infty)$ and sampling it at $k\Delta t$, $k = 0, 1, \ldots, L-1$. This concept is illustrated in Figure 5. (The reason why it is related to the Hankel matrix is that its theoretical foundations are rooted in the Hankel operator theory (see Advantage I), and, from an LTI system perspective, the continuation of the discrete LTI system parameterized by the Hankel matrix $\overline{\mathbf{H}} = \texttt{hankel}(\mathbf{h})$ plays the same role of the continuous LTI system in a canonical SSM in [1].)
>
>    [1] Gu, Albert, Karan Goel, and Christopher Ré. Efficiently modeling long sequences with structured state spaces, International Conference on Learning Representations, 2022.
>
> 2. **Action on sequences of different lengths:** The parameter $\Delta t$ does not explicitly depend on the sequence length $L$ *in the model*. It is a parameter that one can tune when initializing the SSM and can also be trained. However, once $\Delta t$ is determined, it does not depend on the sequence length $L$. Hence, the discrete convolutional kernels associated with the two inputs of lengths $L$ and $2L$ are the same, and consequently, the (first $L$) outputs would be the same. We acknowledge that fixing $n$ and $\Delta t$, as $L \rightarrow \infty$, the model cannot memorize the entire input sequence (neither can a standard SSM do this because of its exponentially decaying memory); the reason why we introduce $\Delta t$ is that it decouples the number of parameters $n$ to the size of the memory, which is now $n / \Delta t$, so for example, if one knows a priori an upper bound of $L$, say $L\_{\max}$, then no matter how large $L\_{\max}$ is, by setting $\Delta t < n / L\_{\max}$, one can guarantee to maintain memory of the entire input sequence --- this cannot be done without $\Delta t$ as in a traditional convolutional model. Although this is not what we used in our model, the idea contained in the reviewer's comment, where the size of the sample period $\Delta t$ also depends on the input length $L$ during the inference time, also sounds very interesting and could be explored in future research.
>
> We hope this answers the reviewer's questions. We would be more than happy to discuss any follow-up questions/thoughts that the reviewer may have.

---

> > ### Comment · Reviewer_TW3G · 2024-11-25
> > **updated score**
> >
> > Thanks for the clarifications. I have updated the score in view of the additional comparisons.

---

> > > ### Author Response · Authors · 2024-11-25
> > >
> > > Thank you for taking our additional clarifications into account and raising your score!

---

### Official Review · Reviewer_1q8g · 2024-10-31

**Soundness:** 2
**Presentation:** 3
**Contribution:** 3
**Rating:** 6
**Confidence:** 3

**Summary:**

This paper develops a new parameterization scheme to help improve training of state space models (SSMs) in sequence modeling. The new parameterization method is based on Hankel operator theory, which resolves the low-rank and numerical instability issues of an LTI system, and the parameterized SSMs could enjoy non-decaying memory. Numerical experiments are conducted on the long range arena benchmark.

**Strengths:**

* The paper is written clearly and the ideas have been presented in the proper order.

* The idea using Hankel operator theory to parameterize state space models is novel, which can provide more insights for other recurrent architectures on sequence modeling.

**Weaknesses:**

* The connections of the Hankel rank/stability and the SSM performance seem not that positively correlated in this paper. For example, from Figure we see that the numerical rank of HOPE-SSM is even lower than init3 before and after training, but these two initializations produce similar numerical performance on the sCIFAR-10 task. Does it mean that even slightly low rank Hankel matrix can achieve good results?

* From Algorithm1, the Hankel operator method looks very similar to the convolution method, it would be better if the authors could add more discussion on these two methods. For example, [1] can be a good reference.

* The notations for $A$ and $\bar{A}$ are not consistently used in this paper. From my understanding, $A$ is the hidden matrix for the continuous system and $\bar{A}$ is the hidden matrix for the discrete system. The HiPPO-Legs initialization is for $A$ matrix, and $\bar{A}$ contains the information of the timescale $\Delta t$. It would be better to distinguish these two notations when making statements. For example, in Figure 4, the perturbation is added to the continuous system while the Hankel singular values are calculated  for the discrete system, and the discrete system contains the information for $\Delta t$. It seems that $\Delta t$ is set to be $1$ in this paper, but in practice, the timescale is scaled based on the sequence length. So what is the reason to take $\Delta t = 1$?
Also, it would be better if the authors could add more details on how the perturbation is added here.

[1] Li, Yuhong, et al. "What Makes Convolutional Models Great on Long Sequence Modeling?." The Eleventh International Conference on Learning Representations.

**Questions:**

* Why do the authors choose to show the Hankel singular values of SSMs trained after 10 epochs instead of after the whole training process (e.g. Figure 3 & 5)?

* How do the authors validate the assumption on the distribution of the diagonal entry of $\bar{A}$ in Theorem 1? For linear recurrent unit [1], $\bar{A}$ is initialized such that $a_j$ is uniformly distributed in a disc $[r_{\min}, r_{\max}]$ where $r_{\min}, r_{\max}$ are both close to $1$.

* Also for Theorem 1, the theoretical bound does not indicate $\beta < 1$, but $\beta \leq 1$ instead. Is there a way to mitigate the logarithmic term in theory?





[1] Orvieto, Antonio, et al. "Resurrecting recurrent neural networks for long sequences." International Conference on Machine Learning. PMLR, 2023.

---

> ### Author Response · Authors · 2024-11-19
> **Response to the Reviewer**
>
> We thank the reviewer for the careful review and insightful comments. Below we address the questions and comments raised in this review.
>
> * W1: The reviewer has made a good observation. We first clarify that after training, HOPE and init3 in fact have about the same amount of large Hankel singular values. The reason why the histogram of HOPE looks denser in the low area is that we used the log-log scale. Hence, even a small number of small Hankel singular values would appear to be significant in the figures. Nevertheless, if one computes the number of relative Hankel singular values $\geq 0.01$, for example, then there are in fact a bit more in the HOPE case than the $\texttt{init}_3$ case (after training). We have made this clarification in section 5. In addition, we admit that the correlation between the singular values and the performance is not totally strict. It does not seem to matter too much if the systems are slightly rank-deficient. The main point here is that seriously rank-deficient LTI systems become the bottlenecks of training good SSMs (i.e., the $\texttt{init}_1$ and $\texttt{init}_2$ cases), which motivates our HOPE model to almost surely avoid degenerate LTI systems.
>
> * W2: The reviewer has raised a good point here. We have expanded section 4 (see the texts in red and Figure 5) to explain this difference in detail and included [Li et al., 2023] as a reference. In particular, HOPE does not parameterize the discrete convolutional kernel (i.e., the discrete impulse response). Instead, it parameterizes a continuous kernel, which will be discretized **with respect to a different $\Delta t$**. That means our method can handle inputs with varying lengths. Moreover, its memory window is not just the past $n$ inputs; instead, its memory window is the past $n / \Delta t$ inputs with a trainable $\Delta t$. We have added an ablation study in Table 3 to indicate the key improvement of HOPE over a standard convolution layer. We hope this elucidates the key difference between our method and a convolutional model.
>
> * W3: Throughout the paper, we use unbarred notations for a continuous-time system and barred notations for a discrete one. We would like to clarify two ideas. (1) In this paper, we only compute the Hankel singular values of continuous LTI systems in an SSM using the MATLAB $\texttt{balreal}$ function. We have made this clear in the caption of Figure 4. (2) In our HOPE model, we only set $\Delta t = 1$ when we (conceptually) identify a continuous LTI system $(\mathbf{A}, \mathbf{B}, \mathbf{C}, \mathbf{D})$ from the Hankel matrix $\overline{\mathbf{H}}$ (i.e., extend a discrete convolutional kernel to a continuous one). When we compute the output of this continuous system, we still discretize it with respect to some trainable $\Delta t$. This has been clarified in section 4 and also see our response to W2 above. We believe Figure 5 is a nice explanation of this intricate idea. We are happy to provide further clarification if necessary.
>
> * Q1: When we did the experiments, we watched the evolution of the singular values throughout training, where we observed that the distributions remained relatively stable after 10 epochs. We apologize for this confusion and have added the histograms after 100 epochs in the appendices (see Figure 10).
>
> * Q2: The statement of Theorem 1 is asymptotic. Assume the eigenvalues are initialized uniformly in the annulus of inner radius $r_{\min}$ and outer radius $r_{\max}$. If we set $r_{\min} < 1$ and $r_{\max} = 1$, then asymptotically we would have $\alpha = 1$. In practice, changing $r_{\min}$ would change the constant in the $\mathcal{O}$-notation. For example, if $r_{\min}$ is very close to $1$, then as $n \rightarrow \infty$, the $\epsilon$-rank is still not proportional to $n$, but one may have to wait until $n$ becomes really large to see that. This explains why the linear RNN in [Orvieto et al.] performs well. If $r_{\max}$ is very close to $1$, but not equals $1$, then one can still apply the theorem with $\alpha = 1$ (though, this would not be tight) to obtain an upper bound of the $\epsilon$-rank.
>
> * Q3: The logarithmic factors come from the Chernoff bound, and would be essentially impossible to eliminate. If we fix some $\alpha > 0$ and $0 < \delta < 1$, however, $\beta$ will eventually be $< 1$ as $n \rightarrow \infty$, which is indeed the case that we consider in Theorem 1. In other words, since the second term in the expression of $\beta$ vanishes as $n \rightarrow \infty$, we could alternatively say that
> \\[
>     \text{rank}\_{\epsilon}(\overline{\Gamma}) = \tilde{\mathcal{O}}(n^{1/(1+\alpha) + \upsilon}) \qquad \text{as } n \rightarrow \infty
> \\]
> for any $\upsilon > 0$, which shows a sublinear growth of the $\epsilon$-rank with respect to $n$.
>
> We hope this answers the reviewer's questions and concerns. We are happy to answer any follow-up question(s) that the reviewer may have.

---

> > ### Comment · Reviewer_1q8g · 2024-11-25
> > **Thank you for your reply**
> >
> > My concerns are addressed, and I will raise my score accordingly.

---

> > > ### Author Response · Authors · 2024-11-25
> > >
> > > We thank the reviewer for confirming that the concerns have been addressed and raising the score!

---

### Official Review · Reviewer_BdQ1 · 2024-11-02

**Soundness:** 4
**Presentation:** 4
**Contribution:** 3
**Rating:** 8
**Confidence:** 3

**Summary:**

This paper proposes a new parametrisation, named HOPE, for Linear Time Invariant (LTI) State Space Models (SSMs), based on the Hankel operator, that improves the performance and makes training more stable. The proposed parametrisation reduces the SSM parameters to 1/3 of the original and works in the frequency domain, hence the Fast Fourier Transform (FFT) and its inverse need to be used during the forward pass. Fortunately, the time and memory complexity of the forward pass remains the same as that of standard LTI SSMs using the convolutional form. The authors first show how current parameterisations can perform poorly both at initialisation or during training and how this can be explained by their corresponding Henkel operator being low-rank (Theorem 1, Figure 3) and the parameterisation being numerically unstable (Theorem 2, Figure 4). Then, they show how their parameterisation effectively mitigates both issues (Theorems 3-4, Figure 5) and does not have an exponentially decaying memory (Advantage III). Each claim Is supported both theoretically and empirically. Empirically, the authors test different parameterisation, initialisation and training schemes on the (noisy-padded) sequential CIFAR dataset, and on the Long Range Arena, also showing how HOPE outperforms previous LTI parameterisations in terms of Test Accuracy.

**Strengths:**

1. Strong empirical and theoretical evidence in favour of the proposed parameterisation.
2. Clearly written.
3. The algorithm is novel, albeit SSM working in the frequency domain have been proposed by (Agarwal et al. 2023).

**Weaknesses:**

1. Some limitations are not experimentally investigated. In particular, the method seems to weigh the input time steps belonging to a fixed window equally and discard the others. This could be a limitation but all tasks considered in the experiments do not seem to benefit from a recency bias (see also the first question).
2. Limited discussion of Related works (Minor).

**Questions:**

1. How does the parameterisation behave in tasks where a recency bias can be beneficial? For example this is the case for generative Language Modelling.
2. Can the authors comment on the relationship between the learned discretization parameter $\Delta_t$ and the size of the window of non-discarded inputs of the HOPE SSM? Is the full input always considered? How does this change during training? Such discussion and potential experimental evidence could strengthen the paper.

Comments:
1. Information on the actual runtime could be moved the main paper since now it is quite hard to find (it is in the last page of the appendix).
2. The paper could benefit from having a discussion on how HOPE differs from spectral SSMs (Agarwal et al.2023) and the stable parameterisation proposed by Wang & Li (2023), which are probably the closest works This could be placed in the appendix. Such discussion could potentially address pros and cons of each approach in addition to the brief discussion in the introduction.

Minor:
- Line 159-161: Can the authors provide a definition of the function space $\ell^2(\mathbb{N})$   and $L^2([0,\infty])$?
- Line 208: imaginal axis -> imaginary axis

---

> ### Author Response · Authors · 2024-11-19
> **Response to the Reviewer**
>
> We thank the reviewer for the careful review and insightful comments. Below we address the questions and comments raised in this review.
>
> * W1: The reviewer has correctly identified an interesting property of the proposed method. Indeed, the current method does not have a natural inductive bias on the time interval $[0,n]$. Suppose recency bias is known to be important in solving a task, we propose the following ways to tune the recency bias:
>
>    1. We can change $\Delta t$ to ask for more or less bias. That is, if $\Delta t$ is set to be larger, then fewer inputs can be fit into the memory window $[0,n]$, so recency bias is advocated.
>
>    2. We can introduce an explicit decay to the Markov parameters. That is, instead of setting $\overline{\mathbf{H}}\_{ij} = \mathbf{h}\_{i+j}$, we use $\overline{\mathbf{H}}\_{ij} = c_{i+j} \mathbf{h}\_{i+j}$ for some $c\_{j} \rightarrow 0$ as $j \rightarrow \infty$. For example, $c\_j = (1+j)^\alpha$ for some hyperparameter $\alpha < 0$. We have discussed these in Appendix L.
>
> * W2: We have added a more detailed comparison between different works in Appendix C.
>
> * Q1: Please see our response to W1. In addition, we have provided some experimental results in Appendix L.
>
> * Q2: $\Delta t$ controls the number of inputs that can be fit into the continuous-time memory window $[0,n]$. Roughly speaking, when computing a certain output $y_k$, the past $n / \Delta t$ inputs will be considered. Hence, if one wants the full input to be considered in computing the last output, then it is required that $\Delta t < n/L$, where $L$ is the length of the sequence. Given that, $\Delta t$ can also be used to tune the recency bias. We have added Figure 5 in the main text to make this parameter more transparent. In addition, Figure 12 in the Appendix shows the effect of changing $\Delta t$ on the system's memory capacity. In practice, we observed that it is much more important to tune a lower bound $\Delta_{\min}$ and an upper bound $\Delta_{\max}$ for $\Delta t$ than to train the parameter.
>
> * C1: It is a nice idea to include more information on the experiments in the main text. We have decided to spend more space to clarify the ideas of HOPE, which, based on the detailed comments from many reviewers, may not have been presented in a crystal clear way. Instead, we directly hyperlinked Table 2 in the main text, which can now be seen by just one click.
>
> * C2: We have added a comparison of different works in Appendix C.
>
> * Minors: We thank the reviewer for the careful read. We have corrected the typo and added the definitions of the $L^2$ and $\ell^2$ spaces.
>
> We hope this answers the reviewer's questions and concerns. We are happy to answer any follow-up question(s) that the reviewer may have.

---

> > ### Comment · Reviewer_BdQ1 · 2024-11-23
> >
> > After having read all the reviews and comments of the authors, I would like to thank and praise the authors for their work during the rebuttal. I think they not only appropriately addressed all of my concerns but I think they did the same for all the reviewers. I especially appreciated the updated version which improved the clarity of the work and my understanding of it, especially by giving a better placement of the work in recent literature. The new experiments on recency bias are quite interesting although preliminary (on a toy problem).
> >
> > Therefore, I confirm my score of 8

---

> > > ### Author Response · Authors · 2024-11-23
> > > **Thank you for your reply**
> > >
> > > We thank the reviewer for reading through our responses and confirming that the questions have been addressed!

---

### Official Review · Reviewer_JzxW · 2024-11-03

**Soundness:** 3
**Presentation:** 3
**Contribution:** 3
**Rating:** 8
**Confidence:** 2

**Summary:**

The paper first examines the stability of SSM training and the recently used initialization method through the lens of Hankel operator theory. The authors argue that by decreasing the effective rank of the Hankel operator, the system loses its expressivity for modeling complex sequences. They then propose a new parametrization of LTI systems based on the Hankel operator, which improves robustness and performance due to the stable spectrum of the Hankel operator during training. Finally, they demonstrate that with this parametrization, memory remains practically constant within a fixed time window. They test this framework on the sCIFAR-10 dataset.

**Strengths:**

1. I believe HOPE is very well motivated by Section 3. The analysis presented in that section clearly illustrates all the issues with current methods based on standard parametrization. In a nutshell, Figures 2 and 3 perfectly summarize the entire section.

2. The three advantages of HOPE are well explained and essential. It is impressive that the authors managed to achieve all three benefits with a simple new parametrization.

3. The entire framework seems very simple, yet powerful.

**Weaknesses:**

Some results lack sufficient motivation. For example, the setup introduced before Theorem 1 could be better explained—why is it reasonable to sample entries of $A$ from $F_a$​ and entries of $\overline{B}\circ\overline{C}^\top$ from a standard normal distribution? Similarly, the discussion preceding Algorithm 1, as well as the algorithm itself, appears more complicated than necessary, with essentially the same information repeated three times. I would prefer having only Algorithm 1 with the full expressions for variables, as in the current prelude to Algorithm 1.

**Questions:**

1. If Mamba achieves SOTA performance, could you explain why there is no experimental comparison with it? If there are no technical limitations and the comparison is meaningful, including it could further enhance the quality of the paper.

2. Could you please explain how Theorem 4 guarantees the numerical stability of HOPE?

3. In Figure 6 (right), why does the memory decay of HOPE still appear significantly smaller than that of S4D after $t=n=64$? Shouldn't HOPE have no memory after $n$?

4. The Hankel matrix looks very similar to the SSM convolution kernel. Could you please explain the differences between the two?

---

> ### Author Response · Authors · 2024-11-19
> **Response to the Reviewer**
>
> We thank the reviewer for the careful review and insightful comments. Below we address the questions and comments raised in this review.
>
> * Weakness: We thank the reviewer for the great structural suggestions. The assumptions of Theorem 1 are mainly due to the range of the variables. The entries of vectors $\mathbf{B}$ and $\mathbf{C}$ are arbitrary real values, so it is natural to assume Gaussian distributions (indeed, this is also usually how we initial them). On the other hand, we have to enforce LTI systems to be asymptotically stable; that is, the eigenvalues of $\overline{\mathbf{A}}$ are contained in the open unit disk. Hence, Gaussian is not an appropriate assumption, but it would be more appropriate to assume properties of the radial distribution instead. (See [1] for example.) We have made this clear before Theorem 1.
>
>    We have also moved the derivation of Algorithm 1 into an appendix and used that space to better conceptually illustrate the idea of HOPE. We believe that the reviewer's suggestion has significantly improved the clarity of our method.
>
>    [1] Antonio Orvieto, Samuel L. Smith, Albert Gu, Anushan Fernando, Caglar Gulcehre, Razvan Pascanu, and Soham De. Resurrecting recurrent neural networks for long sequences. International Conference on Machine Learning, 2023.
>
> * Q1: We consider SSMs and Mambas as models of different purposes. In particular, an SSM is good at modeling the long-range sequences while Mambas have poor performance on the Long-Range Arena benchmark tasks, averaging only a 66.59\% accuracy (see [2]). On the other hand, Mambas have shown great promise in natural language processing, on which SSMs are suboptimal.
>
>    [2] Alonso, C. A., Sieber, J., Zeilinger, M. N., State space models as foundation models: A control theoretic overview, arXiv preprint arXiv:2403.16899, 2024.
>
> * Q2: The numerical stability (or more precisely, conditioning) studies the following question: "Does a small change in the input lead to a big change in the output?" Here, our inputs are the parameters of an SSM ($(\mathbf{A}, \mathbf{B}, \mathbf{C})$ in the S4D case and $\mathbf{h}$ in the HOPE case) and our outputs are the transfer function $G$ of the system. (This means we study the numerical stability of our parameterization of the LTI systems, not that of a particular LTI system.) Theorem 4 says "if we change our inputs $\mathbf{h}$ by a small amount $\Delta$, then the output is changed by at most $\sqrt{n} \Delta$." Hence, the parameterization is relatively stable. This is very different from Theorem 2, where if we change $(\mathbf{A}, \mathbf{B}, \mathbf{C})$ by $\Delta$, the change in the output can be arbitrarily large if $\Lambda(\mathbf{A})$ are close to the imaginary axis.
>
> * Q3: This is an interesting question. The reason why the memory of HOPE does not decay to zero (or the machine precision) is mainly due to the way we compute the impulse response. Given the Hankel matrix $\overline{\mathbf{H}}$ from the trained model, we first identify a discrete LTI system using the MATLAB function $\texttt{imp2ss}$. Then, we take a bilinear transform to obtain the continuous-time LTI system and simulate that system for the impulse response. Errors could arise in three ways, the system identification error, the bilinear transform error, and the discretization error of the continuous-time LTI system. These errors are the main reasons why the impulse response does not go all the way down to machine precision after $t > 64$.
>
> * Q4: The reviewer has correctly identified the similarity between our HOPE-SSM and a convolutional model. We have expanded section 4 (see the texts in red and Figure 5) to explain this difference in detail. HOPE does not parameterize the discrete convolutional kernel (i.e., the discrete impulse response). Instead, it parameterizes a continuous kernel, which will be discretized **with respect to a different $\Delta t$**. That means our method can handle inputs with varying lengths. Moreover, its memory window is not just the past $n$ inputs; instead, its memory window is the past $n / \Delta t$ inputs with a trainable $\Delta t$. We hope this elucidates the key difference between our method and a convolutional model.
>
> We hope this answers the reviewer's questions and concerns. We are happy to answer any follow-up question(s) that the reviewer may have.

---

> > ### Comment · Reviewer_JzxW · 2024-11-25
> >
> > I acknowledge having read the authors' response and I maintain my initial score.

---

> > > ### Author Response · Authors · 2024-11-25
> > >
> > > Thank you for reading through our response and confirming your positive score!

---

### Official Review · Reviewer_iUDX · 2024-11-04

**Soundness:** 2
**Presentation:** 2
**Contribution:** 2
**Rating:** 5
**Confidence:** 3

**Summary:**

This work proposes an alternative parameterization for SSMs using the Hankel operator of the dynamical system, called HOPE. HOPE learns the Hankel operator of a linear dynamical system, instead of directly learning the dynamics matrices as in other SSMs. By using techniques and results in Hankel operator theory, authors show that HOPE is more robust to noise and has long memory. Finally, experiments on LRA show that HOPE is competitive with other SSMs and outperforms baselines on several tasks.

**Strengths:**

This paper proposes a theoretical framework for analyzing SSMs, and there are many supporting graphs and experiments. The plots are well-executed, and the experiments are LRA has a good set of baselines. I also appreciate the noise-padded sCIFAR-10 experiment as an ablation. The algorithm is clearly written.

**Weaknesses:**

1. Motivation: random init in models like Mamba also works, so the initialization issues seem to be limited to certain SSMs.
2. HOPE-SSM does not have non-decaying memory: these are asymptotically stable systems, instead of marginally stable systems, so they cannot have arbitrarily long memory. This point is not clarified upfront in the paper.
3. HOPE-SSM on Path-X, which is the task with the longest memory in LRA, does not outperform S5. This raises some concerns on the long-memory capability of HOPE-SSM in practice.
4. Though authors provide expressivity results on HOPE, there is no guarantee on learning with this parameterization (convergence, etc).

Please also see questions.

**Questions:**

1. In section 3 before 3.1, how should I think about $\epsilon$? Under init3, if $\epsilon=$1e-4, then the frozen model has a higher numerical rank, since there is considerable mass below 1e-4 for the trained model, but the frozen model has worse performance?
2. Why is numerical rank the correct metric instead of the sum of the smallest eigenvalues? ROM quantifies the approximation error using the latter.
3. Theorem 2's upper and lower bounds are off by n, which is significant when the hidden dimension is large (regime of interest for high-order LTIs). In addition, Theorem 4 has a $\sqrt{n}$ on the right hand side, but it's not clear whether the worse $n$ scaling for default SSM parameterization is tight. Can authors comment on this?

---

> ### Author Response · Authors · 2024-11-19
> **Response to the Reviewer (1/2)**
>
> We thank the reviewer for the careful review and insightful comments. Below we address the questions and comments raised in this review.
>
> * W1: We thank the reviewer for raising the interesting problem about the connection between our work and Mamba. In our manuscript, we do not advertise random initialization as an advantage of our HOPE-SSM. The analyses of SSMs and Mambas are essentially different. Mambas rely on time-variant systems, which cannot be studied via the Hankel operator theory. Our paper focuses on SSMs instead, where it has indeed been shown that random initializations are suboptimal (see [1] and [2]).
>
>    [1] Albert Gu, Karan Goel, and Christopher Re. Efficiently modeling long sequences with structured state spaces. In International Conference on Learning Representations, 2022.
>
>    [2] Antonio Orvieto, Samuel L. Smith, Albert Gu, Anushan Fernando, Caglar Gulcehre, Razvan Pascanu, and Soham De. Resurrecting recurrent neural networks for long sequences. International Conference on Machine Learning, 2023.
>
> * W2: We acknowledge that the memory of a continuous-time LTI system in HOPE eventually vanishes. The real practical gain of our model, in terms of its memory, is an empirically non-decaying memory in the early stage. We have made this point clear in the abstract, the introduction (by emphasizing a fixed memory window), and in more detail in section 4 (Advantage III).
>
>    We would also like to remark on an important difference between our method and a convolution-type method that has a fixed memory capacity: our method parameterizes a continuous kernel, which will be discretized **with respect to a different $\Delta t$**. Hence, its memory window is not just the past $n$ inputs; instead, its memory window is the past $n / \Delta t$ inputs with a trainable $\Delta t$ (see Figure 5 that we have added). This makes our model very flexible at decreasing or increasing the memory capacity on a discrete input sequence.
>
> * W3: Our model is the second-best on PathX among all existing ones, and its accuracy is only about $0.1\\%$ lower than that of S5. We believe that this difference is relatively small and does not indicate that our model is necessarily worse than S5 on tasks that require long memory.
>
> * W4: The convergence of HOPE is an interesting problem. We have added a rigorous analysis of the convergence in Appendix J, where we assume a ground truth LTI system that will be learned by HOPE. The settings are very close to those of [3] and [4]. In fact, we showed that the convergence behaviors of HOPE are very nice due to the natural convexity and perfect conditioning of the problem, while the optimization of a canonical LTI system is nonconvex and could possibly be ill-conditioned.
>
>    [3] Jakub Sm\'ekal, Jimmy TH Smith, Michael Kleinman, Dan Biderman, and Scott W Linderman. Towards a theory of learning dynamics in deep state space models. arXiv preprint arXiv:2407.07279, 2024.
>
>    [4] Anonymous authors, Autocorrelation matters: Understanding the role of initialization schemes for state space models, submitted to ICLR 2025.
>
> * Q1: It is true that training the LTI system initialized by $\texttt{init}_3$ makes the Hankel singular values decay faster. This is not a contradiction, though, because although the Hankel singular values decay faster, the additional training of the LTI systems accelerates the optimization in the learning process. They also allow the systems to capture information that could not be learned if the systems are not trained. (See [5] for an example where a fixed LTI system cannot learn high-frequency information.)
>
>    The main message of Figure 3, as opposed to a strict correlation between the singular values and the model accuracy, is that fast-decaying Hankel singular values set bottlenecks for learning the complex input space. For example, the trained systems initialized by $\texttt{init}_1$ and $\texttt{init}_2$ have degenerate LTI systems, essentially preventing a model from doing well. Subsequently, our HOPE-SSM removes this bottleneck by preventing degenerate LTI systems almost surely.
>
>    [5] Anonymous authors, Tuning frequency bias of state space models, submitted to ICLR 2025.

---

> ### Author Response · Authors · 2024-11-19
> **Response to the Reviewer (2/2)**
>
> * Q2: There are a couple of reasons why we chose to study the $\epsilon$-rank instead of the sum of the trailing Hankel singular values:
>
>    1. We have the inequality $\sigma_{k+1} + \cdots + \sigma_{n} \leq (n-k) \sigma_{k+1}$. Hence, as long as $\sigma_{k+1}$ is small, the sum of the trailing singular values is also small (given that we usually use $n \leq 100$ in practice).
>
>    2. The ROM result that we present in the paper controls $\\|G - \tilde{G}\\|_\infty$. However, there are also results that control the physically meaningful $\\|G - \tilde{G}\\|_2$ and $\\|G - \tilde{G}\\|_H$, the Hankel error. Those estimates are based on other forms of singular values; for example. we have $\\|G - \tilde{G}\\|_H \leq \sigma\_{k+1}$. To make our analysis applicable to a broader class of metrics, we choose to study the generic $\epsilon$-rank instead of a specific aggregation of these singular values.
>
> * Q3: The tightness of $n$ is a good question. In Theorem 2(b), our setting is for *any* LTI system to be perturbed. In that case, one cannot show a factor of $n$. However, the $n$ factor in Theorem 2(a) is essentially tight, because fixing any $n$, one can construct an LTI system $\Gamma_n$ of size $n$ and perturbed ones $\tilde{\Gamma}\_{\mathbf{A}}$ and $\tilde{\Gamma}\_{\mathbf{B}}$ so that
> \\[
>     \\|G - \tilde{G}\_\mathbf{A} \\|\_{\infty} \geq n\Delta\_{\mathbf{A}} \max\_j \frac{|b\_jc\_j|}{|\text{Re}(a\_j)|^{2}}, \qquad \\|G - \tilde{G}\_\mathbf{B}\\|\_{\infty} \geq n\Delta\_{\mathbf{B}}\max\_j \frac{1}{|\text{Re}(a\_j)|}.
> \\]
> We have added a proof of this in Appendix F (see Proposition 1). The dichotomy between the factor $n$ and the factor $\sqrt{n}$ from Theorem 2 and 4, respectively, are due to the metric we use to measure the input perturbation. In Theorem 2, we use the $\\|\cdot\\|_\infty$ to measure the input perturbation, while we use the $\\|\cdot\\|_2$ in Theorem 4. We made these choices because they made the results look the cleanest.
>
> We hope this answers the reviewer's questions and concerns. We are happy to answer any follow-up question(s) that the reviewer may have.

---

### Author Response · Authors · 2024-11-19
**Summary of the Rebuttal Revision**

We appreciate the reviewers' thoughtful comments, which have helped improve this manuscript. In addition to the minor changes detailed in our responses to each reviewer, we have made the following key updates (marked in red in the rebuttal version) during the rebuttal period:

* The derivation of Algorithm 1 has been moved to the appendices. This allowed us to use the space in section 4 to further elaborate on the intuition and interpretation of HOPE and clarify its differences from convolution-type models, addressing major concerns about memory issues.

* An ablation study has been added in Table 3 to demonstrate the advantages of HOPE compared to a standard convolutional model.

* We have proven the sharpness of the $n$ factor in Theorem 4(a), which is now included in Appendix F (Proposition 1).

* The convergence of HOPE to a target LTI system using gradient descent has been analyzed in Appendix J (Proposition 2).

* In Appendix L, we have provided guidance on tuning the recency bias in HOPE, along with experimental results.

* A detailed comparison between our model, the Spectral SSM, and the Stable SSM --- both of which address memory issues in SSMs --- has been added to Appendix C.

---

### Meta-Review · Area_Chair_hzfo · 2024-12-19

**Metareview:**

The paper introduces HOPE, a novel parameterization scheme for linear, time-invariant SSMs, leveraging Hankel operator theory to enhance initialization and training stability. The reviewers praised the robust theoretical formulation and strong empirical results demonstrating improved performance on Long-Range Arena asks and sequential CIFAR-10. Strengths highlighted include the insightful use of Hankel operators, non-decaying memory within a fixed time window, and practical improvements over existing HiPPO-based SSMs. The paper is well-executed and addresses critical challenges in training SSMs. Some reviewers raised minor concerns about presentation clarity and specific implementation details. The authors' rebuttal successfully addressed these points, and no major concerns remained post-discussion. Overall, the combination of strong theoretical grounding, novel insights, and empirical improvements makes this a valuable contribution. I recommend acceptance.

**Additional Comments On Reviewer Discussion:**

During the discussion, reviewers highlighted the paper’s strong theoretical contributions and empirical performance on SSMs. Concerns were raised regarding clarity in the presentation and specific implementation details. The authors addressed these points effectively in their rebuttal, providing detailed explanations and additional clarifications where necessary. The reviewers were satisfied with the responses, and no major concerns remained. I considered both the strengths and the resolved concerns, leading to the final decision to recommend acceptance.

---

### Decision · Program_Chairs · 2025-01-22

Accept (Poster)